# MethylResolver—a method for deconvoluting bulk DNA methylation profiles into known and unknown cell contents

Douglas Arneson [1,2 ✉], Xia Yang [1,2,3,4] & Kai Wang [5 ✉]

Bulk tissue DNA methylation profiling has been used to examine epigenetic mechanisms and biomarkers of complex diseases such as cancer. However, heterogeneity of cellular content in tissues complicates result interpretation and utility. In silico deconvolution of cellular fractions from bulk tissue data offers a fast and inexpensive alternative to experimentally measuring such fractions. In this study, we report the design, implementation, and benchmarking of MethylResolver, a Least Trimmed Squares regression-based method for inferring leukocyte subset fractions from methylation profiles of tumor admixtures. Compared to previous approaches MethylResolver is more accurate as unknown cellular content in the mixture increases and is able to resolve tumor purity-scaled immune cell-type fractions without a cancer-specific signature. We also present a pan-cancer deconvolution of TCGA, recapitulating that high eosinophil fraction predicts improved cervical carcinoma survival and identifying elevated B cell fraction as a previously unreported predictor of poor survival for papillary renal cell carcinoma.

[1] Department of Integrative Biology and Physiology, University of California, Los Angeles, Los Angeles, CA 90095, USA. [2] Bioinformatics Interdepartmental Program, University of California, Los Angeles, Los Angeles, CA 90095, USA. [3] Institute for Quantitative and Computational Biosciences, University of California, Los Angeles, Los Angeles, CA 90095, USA. [4] Molecular Biology Institute, University of California, Los Angeles, Los Angeles, CA 90095, USA. [5] Informatics and Predictive Sciences, Bristol-Myers Squibb, San Diego, CA 92121, USA. ✉email: darneson@ucla.edu; Kai.Wang1@bms.com

Complex diseases are the result of pathogenic perturbations in heterogeneous cell populations. For example, infiltrating leukocytes present in the tumor microenvironment (TME) play an important role in cancer progression and patient's survival and response to treatment[1,2]. However, canonical methods for cell composition determination of tissue admixtures, such as flow cytometry and cutting edge single cell technologies like Drop-seq[3], 10X Genomics, and sci-RNA-seq[4], are expensive, labor intensive, require fresh tissues and are sensitive to technical variabilities in cell dissociation procedures. As an alternative, computational methods have emerged to bypass the limitations imposed by experimental approaches and allow for the secondary analysis of troves of publicly available bulk tissue omics data in public data repositories such as GEO and TCGA to derive cellular information.

The seminal method CIBERSORT[5], which is based on a support vector regression method, enables transcriptome-based deconvolution of complex mixtures with unknown content, including solid tumors, using microarrays. However, RNA molecules measured by transcriptome profiling methods are more prone to degradation in realistic clinical applications due to chemical fixation of tissue samples[6], hence present quality issues that likely affect deconvolution results. In contrast, DNA methylation is more stable molecularly[7] and also highly cell-type specific[8], making it an attractive alternative for cellular deconvolution especially in a clinical setting. As such, DNA methylation has been used as a modality for the deconvolution of human whole blood[9–11], and applications of DNA methylation-based deconvolution in the emerging field of immunomethylomics[12] have associated methylation-derived neutrophil-to-lymphocyte ratio (mdNLR) in the peripheral blood as a prognostic factor for survival in cancer[13]. These previous forays into immunomethylomics have primarily been applied to whole blood for which it is easy to obtain a methylation profile for each cell type, but extending this approach into the TME is more difficult as the naïve methods used for deconvoluting whole blood based on well-characterized leukocyte types are not robust to unknown cellular content presented in tumor admixtures.

It is not until recently that methodologies like CIBERSORT were extended as a new method, MethylCIBERSORT[14], to deconvolute the TME using methylation data. MethylCIBERSORT included fibroblasts and relevant cancer cell line profiles in the tumor reference signature matrix, in order to estimate tumor purity (i.e., the percentage of tumor cells in a given sample) and to provide infiltrating leukocyte subset fractions as absolute fractions, rather than relative fractions of each cell type within the leukocyte portion as done by the original CIBERSORT algorithm. Similarly, HEpiDISH[15] included fibroblasts, epithelial cells, and adipocytes into the signature matrix to model additional cell types which might be present in a tumor sample. One of the major limitations of this type of approach is the need to define a signature for every cancer type to infer tumor purity. Further, combining cell-type signatures from different sources without a bridging cell type across datasets may propagate batch effects into the resulting signature. An alternative approach to modeling all possible cell types in a mixture is to generate a clean signature matrix for common cell types of interests (e.g. leukocyte subsets) and use deconvolution approaches that are robust to unknown content which is not modeled in the signature matrix.

One of the motivating factors for developing a deconvolution approach which is agnostic to unknown content is that it can be difficult to identify an appropriate signature for all cancers as each tumor's methylation profile is unique. However, ignoring the tumor contribution to the methylation signature matrix also means that the deconvolution results can at best be presented as relative fractions within leukocyte content rather than absolute fractions in the entire admixture. It has previously been debated that absolute leukocyte subset fractions may have improved prognostic and diagnostic capabilities compared to relative leukocyte subset fractions[16,17]. Estimating absolute leukocyte fractions would require accurate detection of all cell types existing in a tumor admixture, which can be a daunting task due to our limited knowledge of cell composition in the TME of all cancers. Instead, we aimed to provide tumor purity-scaled leukocyte fractions and considered it a step-forward toward estimating absolute leukocyte subset fractions while maintaining a flexible framework.

In a recent publication, the method FARDEEP demonstrated that LTS regression is more robust to outliers and unknown content in the deconvolution of microarray and RNA-seq data compared to other commonly used methods[18]; however, to date no study, to the best of our knowledge, has applied LTS regression for the deconvolution of methylation data. Building upon these findings, we implemented LTS regression for use in deconvolution of DNA methylation data in the form of Methyl-Resolver, a framework for robust deconvolution of the TME using methylation data. In addition to resolving cell-type fractions from bulk methylation data, MethylResolver is able to predict tumor purity without the need of a tumor reference signature. As such, our method can be applied directly to any cancer methylation profile without the need of generating a new reference signature matrix for each different cancer type. We also establish a formal metric to denote the reliability of a deconvolution. Such a metric is often missing or too lenient in other deconvolution approaches leading to spurious deconvolutions in samples without significant presence of known immune content. Here we assemble the most comprehensive reference leukocyte methylation signature to date, consisting of markers of 11 leukocyte cell types, and demonstrate that our method has superior performance in accurately identifying tumor purity from methylation profiles and in deconvoluting tumor-infiltrating leukocyte subset fractions as both relative and tumor purity-scaled fractions without the need for a tumor reference signature. We apply MethylResolver to conduct a pan-cancer deconvolution of TCGA 450k methylation arrays to identify cancer-specific tumor-infiltrating leukocytes most associated with disease prognosis and patient survival, which can be leveraged in the design of targeted immunotherapies.

## Results

**MethylResolver outperformed existing algorithms**. We implemented LTS regression in MethylResolver to conduct deconvolution of tumor admixtures based on DNA methylation profiles to resolve both relative factions and tumor purity-scaled fractions of leukocyte subsets without requiring an input tumor signature. We chose LTS regression due to its ability to learn an optimal subset of CpGs for deconvolution that minimize contaminating signals from unknown content.

We benchmarked the performance of MethylResolver against existing algorithms including LLSR[19], QP[20,21], RLR[22], and nuSVR[5] using both in silico spike-in experiments and in vitro spike-in experiments. These experiments revealed that the LTS algorithm employed in MethylResolver achieved the most robust performance among all methods tested, especially at higher fractions of unknown content (Fig. 1a and Supplementary Fig. 1). The in silico spike-in experiments allow for a head-to-head comparison of all methods with regard to different levels of detection and in the presence of various amounts of unknown content (Methods and Supplementary Fig. 2a). Briefly, in silico mixtures were generated by numerically combining the profiles of six purified immune cell subsets and an epithelial cancer cell line

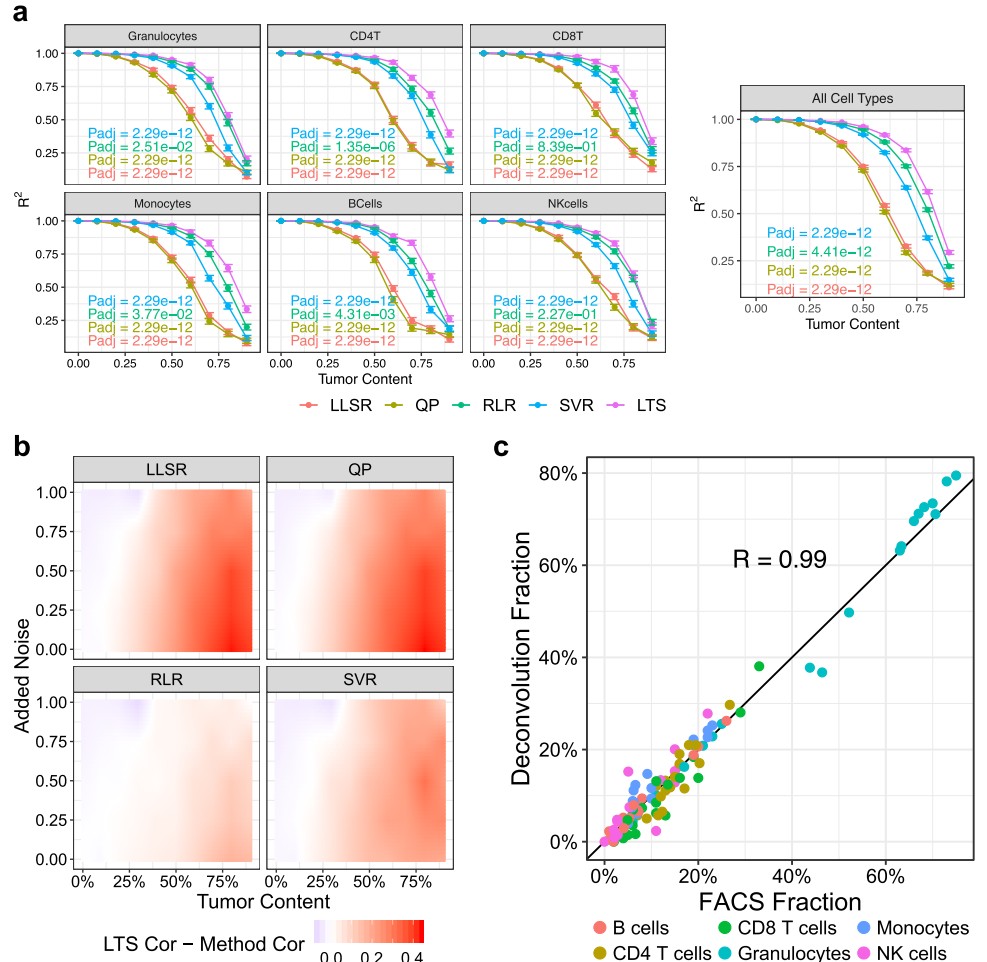

**Fig. 1 Benchmarking MethylResolver. a** Benchmarking five different deconvolution methods using in silico spike-in experiment. Line color corresponds to the deconvolution method, y-axis is the square of the Pearson correlation coefficient ($R^2$) between the inferred cell-type fractions and the ground truth, x-axis is the amount of unknown/tumor content in the mixture, error bars represent SEM. Each panel corresponds to estimates for a specific cell type or the aggregate across all cell types. Statistical significance of the performance of MethylResolver (LTS) over other models was determined using post hoc pairwise comparisons of two-way ANOVA. Adjusted p-values from the ANOVA test are indicated with the color of the text matching the respective model ($n = 90$ per point). **b** Benchmarking four different deconvolution methods versus MethylResolver using the in vitro spike-in experiment. The color in the heatmap corresponds to the difference in the Pearson correlation between MethylResolver and the ground truth and the correlation between the other methods and the ground truth. The y-axis corresponds to the amount of noise that is added, and the x-axis corresponds to the amount of unknown/tumor content in the mixture ($n = 10,800$). **c** MethylResolver predicted relative leukocyte subset fractions (y-axis) of 12 samples from reconstructed mixtures of purified human leukocytes and 6 samples from adult human whole blood with corresponding FACS fractions (x-axis). Cell type is denoted by the color of each point and Pearson Correlation is indicated.

which was absent from the signature matrix and added to comprise 0–90% of the total mixture. As the amount of unknown tumor content added into the mixture increased, the estimated spike-in fractions for the methods that considered all CpGs (LLSR and QP) quickly began to deviate from the ground truth, especially when the spike-in cell type represented less than 5% of the total leukocyte content. RLR and nuSVR were more robust than LLSR and QP, but also began to deviate from the ground truth with higher added tumor content. Across all scenarios, LTS had the highest R-square with the ground truth and performed statistically better than all other methods based on an ANOVA post hoc analysis.

To extend the benchmarking to more realistic scenarios, we conducted an in vitro spike-in experiment using publicly available methylation data on in vitro leukocyte mixtures of known cell-type fractions (Methods and Supplementary Fig. 2b). Briefly, methylation profiling data from experimentally generated in vitro mixtures

were obtained from GSE77797 (Supplementary Data 1), to which a cancer cell line was spiked in at 0–90% to the total mixture, and varying amounts of log-normal noise were also added. Based on the Pearson correlation between the known and estimated cell-type proportions, the LTS regression again outperformed all other deconvolution methods in the in vitro spike-in experiments and was more accurate at high fractions of unknown content (Fig. 1b). Similar results were also obtained when using the absolute Pearson correlation coefficients and root-mean-square error (RMSE) for all benchmarked methods (Supplementary Fig. 3a, b).

LTS regression has a tunable parameter, alpha, that specifies the fraction of variables used in the least-squares regression. We observed similar performance of LTS regression in MethylResolver for a range of alpha values when compared to the other deconvolution methods (Supplementary Fig. 4) and that alpha values between 0.5 and 0.7 had similar performance (Supplementary Fig. 5). LTS regression is essentially a LLSR as alpha approaches

one. To help optimize alpha selection, our MethylResolver implementation empirically tests nine values for alpha from 0.5 to 0.9 in increments of 0.05 and selects the value with the lowest RMSE between the reconstructed immune cell signature profile based on inferred cell fractions and the observed profile, unless users supply their own choice.

**Building a comprehensive immune cell methylation signature**. The IDOL deconvolution reference signature of 6 leukocyte cell types used in benchmarking MethylResolver was designed for deconvolving whole blood. However, this signature misses additional leukocytes which are important in the TME and informative from the standpoint of immunomethylomics and cancer prognosis. To address this need, we built a comprehensive tumor-infiltrating leukocyte methylation signature of 419 CpGs (Supplementary Fig. 6, Supplementary Data 2) containing 11 leukocyte cell types from three separate publicly available studies, after carefully removing the batch effects (see Methods, Supplementary Data 3 and Supplementary Fig. 7).

We first evaluated the performance of this new leukocyte signature matrix on whole blood samples using a set of 18 methylation profiles (GSE77797) with 12 samples from engineered mixtures of purified human leukocytes and six samples from adult human whole blood with corresponding cell-type fractions by FACS (Supplementary Data 4). This evaluation could only be done on six cell types in our extended immune cell signature as there are currently no ground truth fractions with matching methylation profiling data available for the remaining cell types. MethylResolver with our newly defined methylation signature matrix achieved highly accurate cell-type fraction estimates when compared to the ground truth cell-type composition (Fig. 1c and Supplementary Fig. 8a), and similar performance to the original IDOL signature matrix (Supplementary Fig. 8b and Supplementary Fig. 9). We did not expect improvements over previous methods in the deconvolution of whole blood as this is a well-solved problem[9–11]; however, we wanted to ensure the accuracy of our new signature matrix in a task devoid of unknown content. Therefore, our comprehensive leukocyte methylation signature matrix retained the ability to accurately deconvolute leukocyte subset fractions in such mixtures while providing estimates for additional cell types.

**Statistical assessment for the significance of deconvolution**. Not all numerical solutions to a deconvolution problem are reliable given the often-buried immune cell signals in bulk tissue admixtures and the noisy nature of the measurement data. Previous approaches to this problem often lack an assessment of deconvolution result reliability for interpretation. We therefore devised a strategy to assess the statistical significance of a deconvolution by collecting a cohort of true-positive samples from human blood which were expected to have high leukocyte content (Supplementary Data 5) and true negative samples from solid tumor and normal tissue cell lines which were not expected to have significant amounts of leukocytes (Supplementary Data 6). We then explored various goodness-of-fit metrics: R1, R2, RMSE1, and RMSE2, regarding their ability to discriminate between the two sets of samples (details in Methods) as plotted in Fig. 2a and summarized by the receiver operating characteristic (ROC) curves in Fig. 2b. Although all four goodness-of-fit metrics could distinguish true-positive samples from true negative samples with high sensitivity and specificity, the correlation-based metric, R2, which used 210 out of 419 CpGs in our signature matrix determined from LTS regression, performed the best.

To further evaluate the performance of R2 in more realistic complex mixtures, we constructed 200,200 synthetic mixtures by randomly combining pairs of true-positive and true negative samples in various proportions with the true negative percentage ranging from 0 to 100% in increments of 0.1% with 200 samples per increment (Supplementary Fig. 10a, details in Methods). We then applied MethylResolver to calculate the percentage of significant deconvolutions at various R2 thresholds ranging from 0.2 to 0.9 (Fig. 2c, d). This analysis revealed different sensitivity and specificity tradeoffs at different R2 thresholds. At an R2 threshold of 0.5, a true-positive rate of 93.5% at 60% tumor content (which is about the lower bound for typical clinical cancer genomics studies such as those by TCGA) and 0.5% false-positive rate at 100% tumor content were achieved, reflecting high specificity and the ability to avoid low confidence deconvolutions. A more lenient R2 threshold of 0.35 had a false-positive rate of 24.5% at 100% tumor content (Fig. 2c). Users can define an R2 threshold for significant deconvolutions based on acceptable false-positive rates (Fig. 2c).

**Estimating tumor purity-scaled leukocyte fractions**. To generate tumor purity-scaled leukocyte fractions, we hypothesized that the goodness-of-fit metrics from our deconvolution method contain information about the amount of non-leukocyte content in the admixture and could thus be utilized to estimate tumor purity. Indeed, using 7001 samples from TCGA with available tumor purity estimates by CPE (consensus measure of purity estimates) from Aran et al.[23], we observed strong correlations between CPE and each of the four goodness-of-fit metrics (Fig. 2e), as well as relative fractions of certain immune cell types (e.g. Eosinophils, Memory T Cells, and Natural Killer Cells) inferred from our MethylResolver deconvolution (Supplementary Fig. 11). The correlation between CPE and the goodness-of-fit metrics is intuitive as the latter measures how well the reconstructed immune cell signature profile using inferred immune cell fractions recapitulates the observed profile. As tumor purity increases, the inference of non-tumor immune cell fractions becomes more difficult and the reconstructed profile will deviate further from the observed profile.

Motivated by these observations, we trained a random forest (RF) model using TCGA samples that predicts CPE-based tumor purity using features from our LTS deconvolution (Supplementary Fig. 10b, details in Methods). Briefly, the RF model was trained and tested on half of the samples from each cancer type in TCGA, and then evaluated on the other half of the samples which were completely held out from the model training. The variable importance of each feature highlighted the importance of the goodness-of-fit metrics RMSE2, R2, RMSE1, and the relative fractions of Eosinophils and Memory T Cells for estimating tumor purity (Supplementary Fig. 12). When evaluated on the held-out samples using Pearson correlation between predicted tumor purity values by our approach and those from CPE (Fig. 3), our RF model achieved a high correlation ($0.72 < r < 0.95$) across all cancer types. To attain performance estimates for tumor purity inference for cancer types not seen in the training, we sequentially held each cancer type out entirely from the training of the model and evaluated the performance on the held-out cancer types (Supplementary Fig. 13). Most cancer types achieved a correlation $r > 0.75$ in the leave-one-out analysis indicating a lower bound on tumor purity estimates for cancer types not used in RF model training. Furthermore, our RF regression model was able to predict the tumor content within 10% of the CPE purity estimate for 89% of the samples (Supplementary Fig. 14), demonstrating its accuracy.

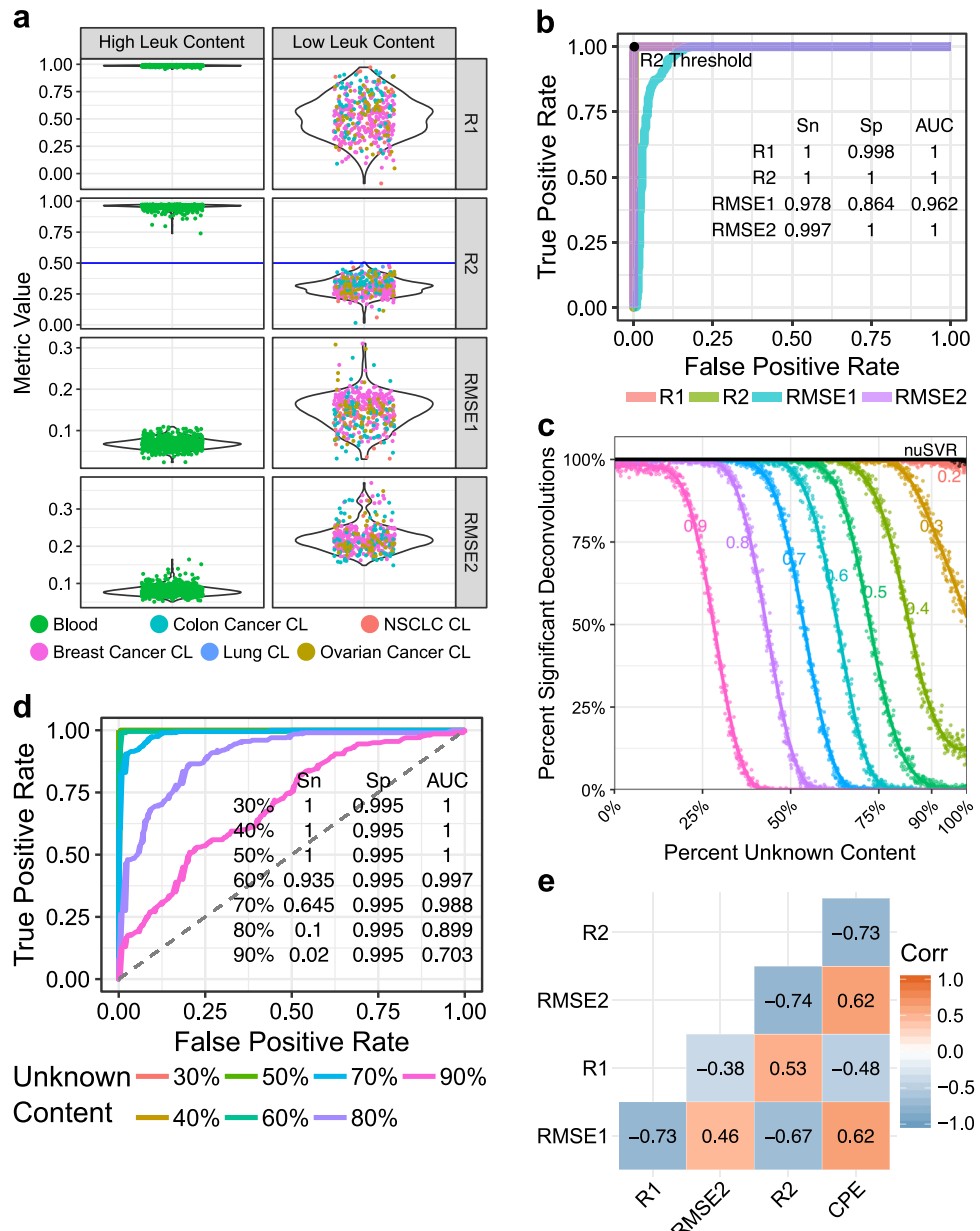

**Fig. 2 Determining a threshold for significant deconvolutions and identifying tumor purity to resolve tumor purity-scaled leukocyte subset fractions. a** Four goodness-of-fit metrics considered for determining significant deconvolutions (rows) are benchmarked for their abilities to stratify samples with putative high and low leukocyte content. The score for each metric is given on the *y*-axis and different sample types are indicated by color. The R2 threshold used to determine significance of deconvolution in this work (blue line) is indicated. **b** ROC curves demonstrating the of the ability of the four goodness-of-fit metrics to stratify positive and negative cohorts. The sensitivity and specificity of these metrics at the point which gives the highest Youden's J statistic is indicated along with the location of the R2 significance threshold. **c** A range of R2 thresholds from 0.2 to 0.9 (colored numbers) are tested for their ability to call significant deconvolutions (y-axis) of synthetic mixtures of varying fractions of unknown content from 0 to 100% in increments of 0.1% with 200 random synthetic mixtures at each increment (*x*-axis). The performance of CIBERSORT (nuSVR) on the same mixtures using *p*-values obtained from 2500 permutations is also indicated (black line). **d** ROC curves demonstrating the performance of the R2 threshold to significantly deconvolve mixtures at different percentages of unknown content. Sensitivity and specificity shown for R2 = 0.5 ($n = 200,200$). **e** Correlations between CPE tumor purity estimates and the four goodness-of-fit metrics for 7001 samples from TCGA.

**Pan-cancer analysis of TCGA.** We applied MethylResolver to the methylation profiling data of 9,756 TCGA tumor samples across 33 different cancer types in 11 broad categories. Different R2 thresholds for significant deconvolutions can be chosen based on acceptable false-positive and false-negative rates in our simulation study (Fig. 2c). A more lenient R2 threshold of 0.35 had a false-positive rate of 24.5% (Fig. 2c) and resulted in significant deconvolutions for

a majority (75.8%) of the TCGA tumor mixtures (Fig. 4a). A more stringent R2 threshold of 0.5 had a false-positive rate of 0.5% (Fig. 2c) and resulted in fewer (35.2%) significant deconvolutions of TCGA tumor mixtures (Fig. 4b). We used the latter R2 threshold for all downstream analyses for its better balance between false positives and false negatives predictions. For samples with significant deconvolutions within each cancer type, sample-level

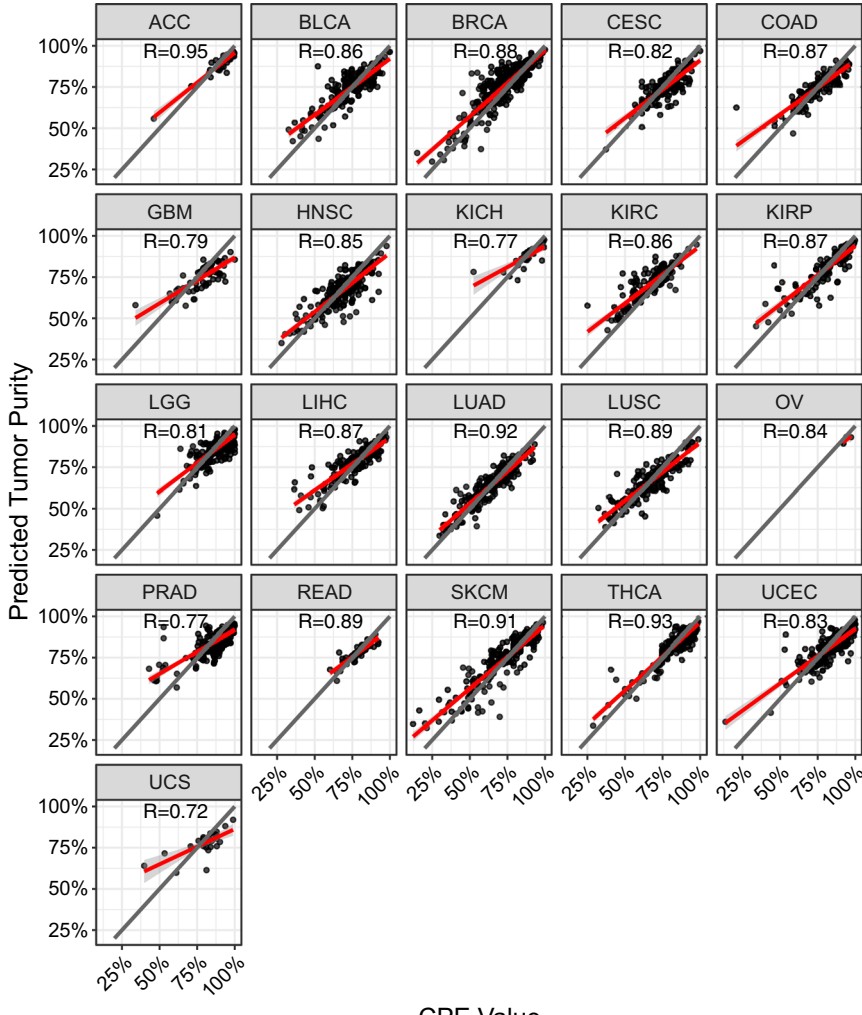

**Fig. 3 Performance of MethylResolver tumor purity estimation.** Correlation between MethylResolver predicted tumor purity from our RF regression model (y-axis) and the ground truth CPE tumor purity value (x-axis) for 21 different cancer types from TCGA (panels) with the Pearson correlation indicated. The gray line is y=x and the red line is a linear regression of the data points. The RF regression model was trained on half the samples from each cancer type and the cancer samples displayed here were held out from the training of the model (n = 3497).

(Supplementary Data 7 and Supplementary Data 8), as well as the average relative (Fig. 4c) and tumor purity-scaled (Fig. 4d) fractions of the 11 different cell types in our reference signature were calculated. Tumor purity-scaled leukocyte subset fractions were not inferred for hematologic cancers. For these cancers, our results recapitulated the immune cell lineages from which they originate, e.g., high fractions of naïve T cells in Thymomas (THYM)[24], B cells in diffuse large B-cell lymphoma (DLBCL)[25], and monocytes in Acute Myeloid Leukemia (LAML) (Fig. 4c), lending support to the validity of our approach.

For each cancer type, we also correlated both the relative (Supplementary Data 9) and tumor purity-scaled (Supplementary Data 10) cell fractions at the sample level with known metrics representative of cytotoxic T cells and natural killer cells, such as the 10-gene IFN-γ score (*IFNG, STAT1, CCR5, CXCL9, CXCL10, CXCL11, IDO1, PRF1, GZMA*, and *MHCII HLA-DRA*)[26] and a cytolytic activity score (average expression of *PRF1* and *GZMA*)[27]. We observed strong Spearman correlations between cytolytic score and IFN-γ score, and the combined tumor purity-scaled fractions of CD8 T cells and NK cells in many cancer types (Fig. 5a). Given that these scores are based on gene expression values determined by RNA sequencing whereas the deconvolution results from

MethylResolver are based on methylation arrays, these strong correlations between the MethylResolver-derived tumor purity-scaled fractions of cytotoxic T cells and multiple metrics known to be associated with these cell types from real biological samples further supports the accuracy of our method.

**Identification of prognostic leukocyte subsets in TME.** We leveraged both the tumor purity-scaled leukocyte subset fractions (Fig. 5b–e) and the relative fractions (Supplementary Fig. 15) derived from MethylResolver to screen for leukocyte subsets that are prognostic of patient survival. We employed rigorous statistical practices for identifying prognostic leukocyte subsets including using a stringent threshold for significant deconvolutions, a regression framework which accounted for important confounders, and multiple testing correction. While this rigor resulted in fewer significant associations compared to other similar works, we were confident in their validity.

Through this analysis, we were able to identify known and previously unreported leukocyte populations with significant impact on prognosis for many of the cancer types in TCGA (Fig. 5b). For example, we confirmed that higher tumor purity-

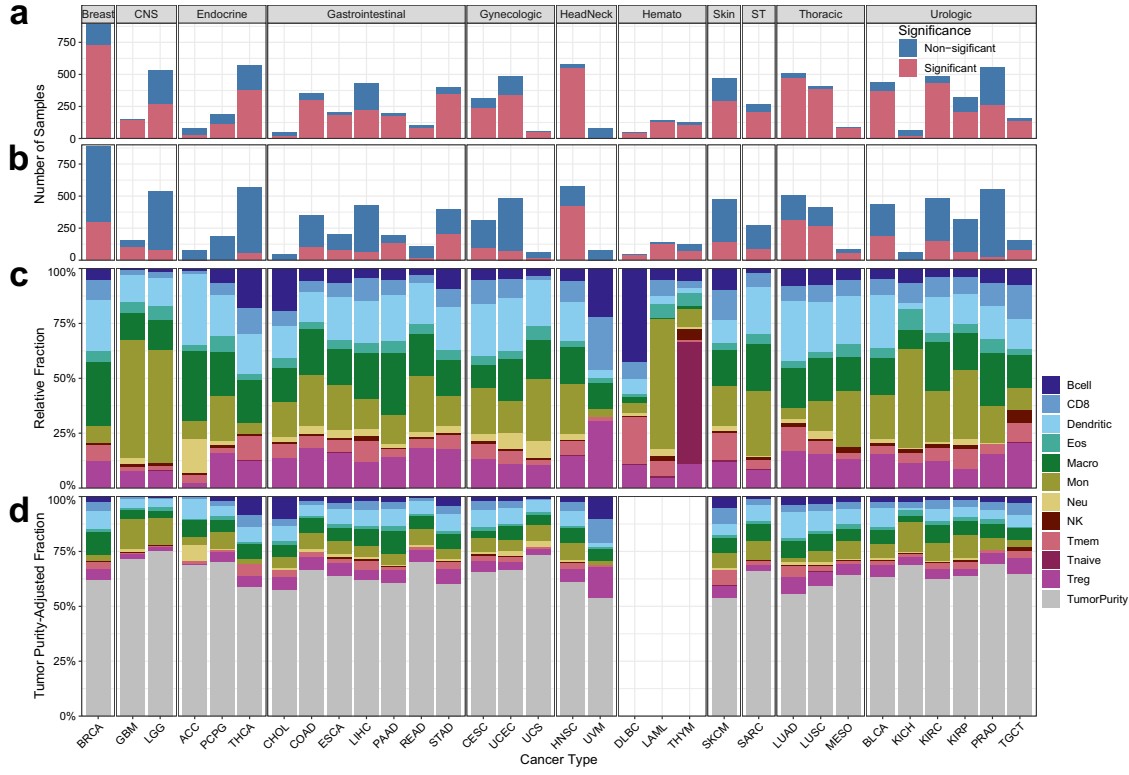

**Fig. 4 Pan-cancer deconvolution of TCGA. a–d** MethylResolver was applied to 9,756 samples from 33 cancer types from TCGA group into 11 broad categories. The total number of samples profiled per cancer and the fraction of samples which had a significant deconvolution (red) with a loose (**a**) and stringent (**b**) statistical threshold. **c** Relative and (**d**) tumor purity-scaled leukocyte subset fractions for the significantly deconvoluted TCGA samples with cell type indicated by color. Tumor purity-scaled leukocyte subset fractions are not inferred for hematologic cancers.

scaled eosinophil fraction in the TME is predictive of better survival outcomes for Cervical Squamous Cell Carcinoma (CESC) (Fig. 5b, c)[28,29]. Among the new findings we predicted that higher B cell fraction in the TME was predictive of worse survival outcome for kidney renal papillary cell carcinoma (KIRP) (Fig. 5b, d). We also found a higher tumor purity-scaled B cell fraction in the TME to be predictive of improved survival outcome for Pancreatic Adenocarcinoma (PAAD) (Fig. 5b, e).

In general, we observed broadly consistent patterns between the prognostic associations generated from the tumor purity-scaled and relative leukocyte subset fractions (agreeing on 21 significant associations out of 25 and 27). However, some of the unique cases found only when using tumor purity-scale leukocyte subsets are supported by literature including improved prognosis due to tissue associate eosinophilia (TATE) in cervical cancers (CESC)[28] and better survival outcome in pancreatic cancer (PAAD) treated with chemoradiation with high eosinophil-to-lymphocyte ratio (ELR)[30].

## Discussion

To facilitate DNA methylation-based leukocyte deconvolution for diagnostic and therapeutic applications, we present a new LTS-based robust deconvolution method, MethylResolver, which uses reference leukocyte cell methylation signatures to provide both relative and tumor purity-scaled leukocyte subset fractions from bulk tumor methylation profiles. Compared to previous methods, MethylResolver has the following advantages: (1) it introduces a robust method for the deconvolution of bulk tissue methylation profiles and has superior performance compared to previous methods; (2) beyond relative leukocyte subset fractions, it is also capable of inferring overall tumor purity and tumor purity-scaled

leukocyte subset fractions; (3) it provides the most comprehensive leukocyte subset methylation signatures to date; (4) it offers a way to assess the significance of a deconvolution.

The unique feature of MethylResolver to accurately infer immune cell subset fractions in the tumor tissue admixture without requiring a signature for the unknown content or having to generate the reference signature matrix for each cancer is a key advantage of MethylResolver over previous reference-based deconvolution methods like MethylCIBERSORT[14]. MethylResolver's ability to infer tumor purity measure also allowed us to produce not only relative leukocyte subset fractions from tumor admixtures, but also tumor purity-scaled fractions. Tumor purity-scaled fractions represent a step towards absolute fractions which may offer improved prognostic and diagnostic capabilities compared to relative leukocyte subset fractions[16,17]. Indeed, we identified unique prognostic biomarkers using tumor purity-scaled fractions that are supported by literature. We note that tumor purity-scaled fractions are not absolute fractions as we do not currently estimate the contributions of other cell types existing in the TME (e.g. fibroblasts and endothelial cells). Nevertheless, the purity-scaled fractions are closer to the absolute fractions than relative fractions.

The introduction of deconvolution significance is another major contribution of MethylResolver. All reference-based deconvolution methods will mathematically decompose a mixture into some set of coefficients under the assumption that they represent the fractions of cell types presumably present in the reference signature matrix. However, in situations where a cell-type is truly not present, or the amount of unknown content is so overwhelming that an accurate deconvolution becomes infeasible, a statistical assessment is required to discriminate these unreliable deconvolutions from

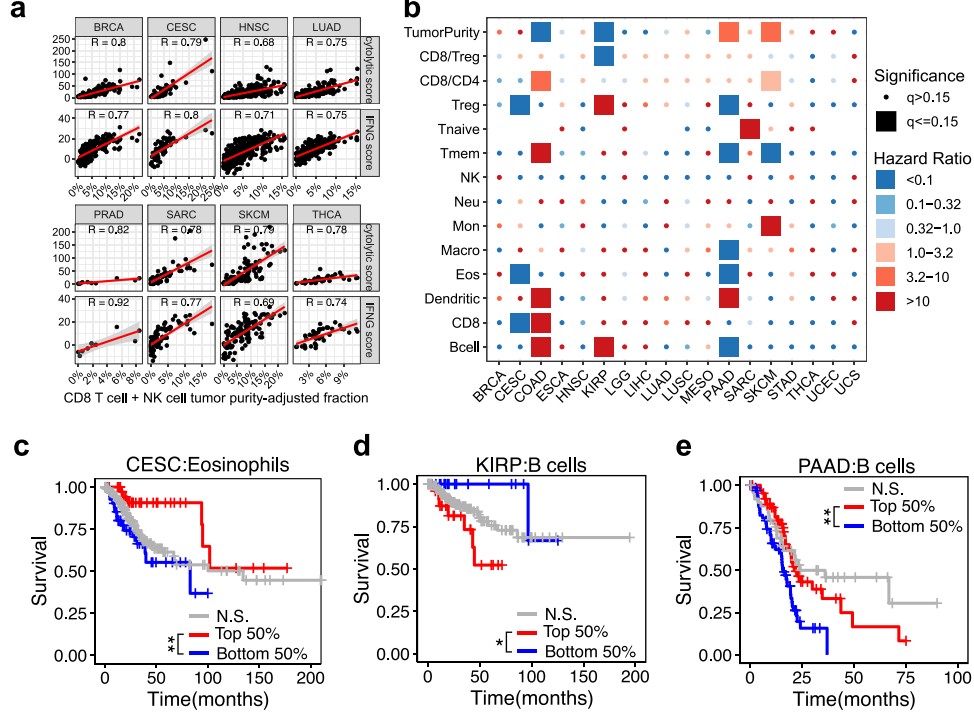

**Fig. 5 Prognostic potential of tumor purity-scaled leukocyte subset fractions. a** Spearman correlation of the MethylResolver-derived tumor purity-scaled fraction of CD8 T cells + NK cells with genes and scores known to correlate with these cell types profiled across eight cancer types. The *y*-axis is the score or gene expression (in FPKM) and the *x*-axis is the tumor purity-scaled fraction of the CD8 T cells + NK cells. The red line is the linear regression fit of the points. **b** Cox regression was applied to the MethylResolver pan-cancer deconvolution of TCGA to infer prognostic leukocyte subsets using tumor purity-scaled fractions from significant deconvolutions. Heatmap colors correspond to the hazard ratio values and shapes correspond to the significance, rows correspond to cancer type and columns correspond to cell-type, tumor purity, CD8-to-Treg ratio (CD8/Treg) or CD8-to-CD4 ratio (CD8/CD4). Only samples with significant deconvolutions were used in the Cox regression. **c**–**e** Kaplan–Meier plots showing patients' overall survival stratified by median tumor purity-scaled fraction of eosinophils in CESC (**c**), median tumor purity-scaled fraction of B cells in KIRP (**d**), median tumor purity-scaled fraction of B cells in PAAD (**e**). Red lines show survival of the top 50% tumor purity-scaled fractions of the indicated cell-type/feature from significant deconvolutions, blue lines show the survival from the bottom 50% from significant deconvolutions, and gray lines show the survival from non-significant deconvolutions. *q < 0.05 and **q < 0.01.

trustworthy deconvolutions. The significance threshold provides users with a confidence level that avoids overinterpretation of the results, however, it does not limit the use of the deconvolution results. The threshold can be thought of as analogous to an FDR assessment which provides a metric to quantify the trustworthiness of results but also allows for investigation of said results if they do not meet a pre-defined cutoff.

Using a stringent deconvolution significance threshold, 3430 (35.2%) of the TCGA tumors analyzed yielded a significant deconvolution. This is not surprising given that the TCGA selection of tumor samples favors samples with high tumor purity, resulting in a smaller fraction of leukocytes and other cell types in a majority of the samples. Our findings that DLBC, LAML, HNSC, PAAD, and MESO (Fig. 4b) have the highest percentage of significant deconvolutions largely agree with the ranking of cancers with highest leukocyte factions generated by Thorsson et al. as well the fact that these cancers are most responsive to immune-checkpoint inhibitors[31]. In contrast, tumor types such as PRAD had the lowest percentage of significant deconvolutions (Fig. 4b), which is consistent with the general belief that prostate cancer is often considered as an "immune desert" with a low leukocyte fraction relative to stromal content, and responding poorly to immune-checkpoint-pathway inhibitors[31–34]. In such cases, the immune cell subset fraction estimates by deconvolution would not be trustworthy and looking at the prognostic value of such estimates could be misleading.

Therefore, our deconvolution significance measure is biological meaningful and relevant. However, if a higher false-positive rate is acceptable the significance threshold can be tuned to allow for the deconvolution of tumor samples with higher fractions of unknown content (Fig. 4a).

Application of MethylResolver to a pan-cancer deconvolution of the TCGA dataset also demonstrated its ability to capture known biology. For instance, high B cell content was found in B cell lymphoma[25] (Fig. 4c, d), tumor purity-scaled fractions of CD8 T cells + NK cells were correlated with cytolytic score[27] and IFN-γ score[26] (Fig. 5a), and high eosinophil content was predictive of better survival in cervical squamous cell carcinoma (CESC)[29] (Fig. 5b, c). This latter finding is supported by prior evidence that moderate and intense tumor-associated tissue eosinophilia (TATE) (>30 eosinophils/mm²) was associated with significantly improved 5-year survival prognosis[28]. Additionally, it has been reported that radiotherapy of CESC patients leads to increased blood absolute eosinophil count (AEC) and that individuals who responded well to the radiotherapy had a larger increase in AEC and TATE than the poor responders, thus indicating that eosinophils are predictive of radiation response in CESC patients[29]. These results give us the confidence to extend the method to glean potentially novel insights in the prognostic impact of various leukocyte populations on different cancer types.

Among the previously unreported findings, we found that higher B cell fraction in the TME was most predictive of worse survival

outcome for kidney renal papillary cell carcinoma (KIRP) (Fig. 5b, d). Although there were no prior prognostic studies of B cells for KIRP, high infiltration of B cells in renal cell cancer (RCC) and kidney clear cell carcinoma (KIRC) has been associated with poor prognosis for these cancers[35]. Mechanistically, it has been proposed that IL-10-producing B cells can lead to T cell immunosuppression in renal cell carcinoma[36] which can potentially explain the prognostic effect seen using our MethylResolver deconvolution of KIRP. We also predicted a higher tumor purity-scaled B cell fraction in the TME to be predictive of improved survival outcome for Pancreatic Adenocarcinoma (PAAD) (Fig. 5b, e). This is contrary to a previous study that indicates tumor-infiltrating B lymphocytes showed a non-significant trend for worse prognosis, however, the same study also demonstrated that B cell tertiary lymphoid tissue (TLT) were associated with significantly longer survival[37]. We hypothesized that MethylResolver may have detected the high density of B cells in these peri- and intratumoral TLT structures, known to be associated with improved prognosis in many cancer types including pancreatic cancer[38], which is driving the underlying association of higher tumor purity-scaled B cell fractions with improved prognosis. Although we identify fewer leukocyte prognostic associations than previous works, our rigorous statistics and multiple testing correction give us confidence in these findings, however, future validation is required.

While MethylResolver has made a number of improvements over existing methodologies, we acknowledge that MethylResolver has the following limitations. First, the TME includes not only leukocytes but also other cell types such as stromal cells, endothelial cells, and other tissue specific components which have been modeled in methods like HEpiDISH[15] and MethylCIBERSORT[14] to attain absolute leukocyte subset fractions. However, to model every cell type which differs between tissues can be difficult. To avoid this need, we employed a methodology that is more generally applicable, where we built a single signature matrix of the common cell types of interest in TME (i.e. immune cells) and devised an algorithm that is robust to the unknown content (i.e. cell types in TME that are not modeled). In future extensions of this work we would be very interested in including additional cell types such as endothelial cells when appropriate data with overlapping cell types to facilitate batch correction become available. Second, although we have constructed the most comprehensive leukocyte subset methylation signature matrix to date, it is far from exhaustive and does not include subpopulations of cells like CD4+T cell subsets (e.g. Th1, Th2, Th17) which have been previously reported to have different associations with patient prognosis[39]. Future efforts in purifying and profiling specific immune cell subpopulations, or single cell sequencing would greatly improve the ability to generate a higher resolution signature matrix. Moreover, although we are able to demonstrate that our tumor purity estimates are comparable to CPE tumor purity estimates, the accuracy of this metric needs to be further benchmarked against a true gold standard (e.g. known tumor purities measures), which is currently challenging to obtain. Lastly, the new observations reported in this study require further testing and validation.

In summary, we developed MethylResolver, a robust reference-based deconvolution method which can provide tumor purity-scaled fractions of tumor-infiltrating leukocyte subsets in the TME using DNA methylation profiles of bulk tumor samples. For immediate applications, we expect MethylResolver can facilitate the efforts in understanding cancer biology and in identifying biomarkers for cancer immunotherapy. In addition, as cell-type composition is a known confounder in both epigenome-wide association studies and in differential methylation studies[9,40–42], MethylResolver is an intuitive application that can be used to provide cell-type fraction estimates to correct for cell composition in such studies. When high-throughput single cell methylation data becomes readily available, we also envision that MethylResolver could be extended to generate de novo cell-type specific methylation signature matrices similar to recent single cell RNA-Seq based approaches such as MuSiC[43], Bisque[44], and CIBER-SORTx[45]. Even at its current stage, we believe MethylResolver is a valuable resource and addition to the field of immunomethylomics and in silico deconvolution.

## Methods

**Measurement of methylation levels.** Methylation data are typically reported as Beta values bounded between 0 (unmethylated) and 1 (fully methylated). We select methylation features with low levels of DNA methylation, which likely translate to high cell-type-specific gene expression[46,47].

**Statistical methods for deconvolution.** There are a number of different reference-based methods which have previously been employed to deconvolve methylation mixture profiles, both with[5,14] and without[10,11] unknown content, and a number of reference-free deconvolution methods[42,48]. Due to limitations in accuracy and biological interpretability of underlying cell types of reference-free deconvolution methods, here we focused on the reference-based methods. Most of the reference-based methods utilize a similar framework where a methylation mixture, $m$, is represented as a system of equations. Given a pre-defined cell-type signature matrix, $B$, with each column representing a cell type and each row a different CpG, we solve the fractions of each cell type, $f$, using the equation $m = B \times f$. The main differences among the methods are the underlying statistics. Previously Linear Least Squares Regression or LLSR[19], Quadratic Programming or QP[20,21], Robust Linear Regression or RLR[5], and nu Support Vector Machine Regression or nuSVR[5], have been used. Here we leveraged a new statistical approach, Least Trimmed Squares Regression or LTS[49,50], for robust reference-based deconvolution of methylation data, and compared it with the previously used statistical methods.

For LLSR, the least-squares fit for the system of linear equations specifying $m = B \times f$ was derived with the 'lsfit' function in R. The constraint of non-negative numbers was met by removing the lowest negative coefficient from the fit equation and iterating until all coefficients were non-negative. Cell type fractions were scaled to sum to one.

For QP, the "lsqlin" function from the R package "pracma v1.9.9" was used to solve a linearly constrained linear least-squares problem by finding the global optimal solution which minimized the residuals of the least squares given a non-negative constraint[51]. Cell type fractions were scaled to sum to one.

For RLR, the "rlm" function in the R package "MASS" with M-estimation and Huber weighting was used. In M-estimation, the weight function defines a co-dependence between the residuals and the weights, which is solved using Iteratively Reweighted Least Squares (IRLS). Huber weighting results in observations with small residuals having a weight of 1 and larger residuals with weights that decrease as the residual increases. This effectively puts more weight on the CpGs which best explain the system of equations and less weight on those that cannot.

The CIBERSORT framework was based on nuSVR[52] and the R source code (v1.04) was obtained from https://cibersort.stanford.edu. nuSVR was implemented with the "svm" function in the R package "e1071 v1.7-0". nuSVR performs a regression by discovering the hyperplane which fits as many of the data points as possible within a given distance ε from the hyperplane. Points which are farther from the hyperplane than ε are evaluated with the loss function. In addition to minimizing the loss function, nuSVR also seeks to minimize the penalty function which penalizes model complexity. The *nu* parameter serves as an upper bound on the training errors and a lower bound on the fraction of support vectors, in this case CpGs. The CIBERSORT implementation used three different values for *nu*: 0.25, 0.5 and 0.75, corresponding to fitting an nuSVR to 25%, 50%, and 75%, respectively, of the CpGs in the signature matrix and chose the model which had the smallest residual error. Negative coefficients were set to zero and then all coefficients were normalized to sum to 1.

For MethylResover, LTS regression was designed to fit data while minimizing the effect of outliers. These outliers may come from CpGs in the signature matrix that also bear signal in the unknown content present in the mixture to be deconvoluted. LTS regression was implemented with the "ltsReg" function in the R package "robustbase v0.93-2". LTS regression is similar to LLSR where it seeks to minimize the sum of squared residuals; however, LLSR computes this quantity across all CpGs, whereas LTS finds and uses the optimal CpG subset of specified length. Our implementation considered nine values for alpha: from 0.5–0.9 in increments of 0.05 corresponding to fitting a regression to 50–90%, of the CpGs in the signature matrix and chose the model with the smallest RMSE of the reconstructed signature profile based on inferred cell-type fractions compared to the observed profile.

**Model formulation.** Deconvolution can be represented as a linear model,

$$m = B \times f,$$

where $m \in \mathbb{R}^n$ is the Beta value for $n$ CpGs in the methylation mixture, $B \in \mathbb{R}^{n \times m}$ is the signature matrix which has the mean Beta values for $m$ cell types, and $f \in \mathbb{R}^m$ are the unknown cell-type fractions. The cell fractions can be estimated with ordinary least squares (OLS),

$$\hat{f}_{ols} = \left(B^T B\right)^{-1} B^T m,$$

where $\hat{f}_{ols}$ are the estimated parameter values. The residuals $r = (r_1, \ldots, r_n)$ can be obtained from the OLS estimate,

$$r = m - B\hat{f}_{ols}.$$

In standard least squares the parameter estimates are obtained by minimizing the sum of squared residuals,

$$S\left(\hat{f}_{ols}\right) = \sum_{j=1}^{n} r_{(j)}\left(\hat{f}_{ols}\right)^2,$$

where the sum of squared residuals is computed across $n$ CpGs. LTS iterates through subset CpGs of size $k$ from the full set of $n$ signature matrix CpGs to attempt to find the $k$ subset that yields the lowest sum of squared residuals,

$$S_k\left(\hat{f}_{ols}\right) = \sum_{j=1}^{k} r_{(j)}\left(\hat{f}_{ols}\right)^2,$$

where the $n$-$k$ CpGs do not influence the fit. The size of $k$ is specified by the $\alpha$-parameter, where $1 \le \alpha \ge 0.5$ and $k = n \times \alpha$. The $k$ subset CpGs are obtained using the Fast LTS algorithm[53]. By subsetting to the $k$ set of CpGs which minimize the sum of squared residuals, LTS can remove CpGs that bear signals from unknown content in the mixture not modeled in the signature matrix, thereby improving the cell fraction estimation of the modeled cell types.

**Benchmarking with in silico spike-in experiments**. As a baseline benchmark to assess the detection limit of each method for rare cell types in bulk tissues, we first performed an in silico spike-in experiment using the pre-established IDOL (**Id**entifying **O**ptimal **L**ibraries) whole blood reference DNA methylation signature[11], as this signature has been previously used to benchmark other methods. Here, six purified immune cell profiles, which corresponded to the cell types in the IDOL signature, were taken from a publicly available dataset (GSE88824) and were mixed in varying proportions with tumor content (in the form of HCT-116, 451-LU, or MCF-7 epithelial cancer cell lines) to generate an in silico methylation profile of immune cell and tumor cell mixtures. For each in silico mixture, we spiked in tumor content ranging from 0 to 90% with the remaining mixture fraction of the methylation sample comprised of the six immune cell types. Of this remaining fraction, one of the six immune cell types from the signature matrix was spiked in at a certain percentage of the total immune cell content (from 0.5 to 50%) with the rest of the amount randomly distributed across the five other immune cell types. Statistical significance of the performance between the five deconvolution models was determined using post hoc pairwise comparisons of two-way ANOVA. An ANOVA model was fit with the 'aov' function in R treating different deconvolution methods and the tumor content composition as categorical independent variables, and the predicted fraction as the dependent variable. This was done for each cell type separately and the adjusted p-values were computed using Tukey's "Honest Significant Difference" method with the "TukeyHSD" function in R. The purpose of this in silico spike-in experiment was to create a fair comparison to all methods with regard to different levels of detection and in the presence of various amount of unknown content, rather than evaluating their real-world performance. The latter was assessed in the subsequent experiments.

**Benchmarking with in vitro spike-in experiments**. To benchmark and evaluate the performance of each deconvolution method in a more realistic context, we performed further benchmarking experiments using 12 experimentally generated in vitro mixtures obtained from GSE77797. Specifically, these mixtures were created by experimentally mixing six immune cell types in known proportions to obtain DNA mixtures, followed by methylation profiling on an Illumina HumanMethylation450 BeadChip. Similar to the above in silico benchmarking, an unknown tumor content (in the form of HCT-116, 451-LU, or MCF-7 epithelial cancer cell lines) was added to the in vitro mixtures (from 0 to 90% in 10% increments) as well as a log-normal noise in the form of $2^{N(0, \lambda \sigma)}$ ($\lambda = 0$-1 in 0.1 increments). Each method was applied to the in vitro spike-in mixtures to test its performance in recovering the known proportions.

**Building a more comprehensive methylation signature matrix**. The IDOL signature used in the benchmarking analyses only contained 6 leukocyte cell types. To build a more comprehensive leukocyte methylation signature, we gathered methylation data for 11 cell types from three separate publicly available studies (GSE35069, GSE59250, GSE71837). Visualization of the 269 samples from these datasets by embedding of the top 20 principal components in a UMAP (Uniform Manifold Approximation and Projection)[54] plot using the R package "umap" revealed segregation by dataset (Supplementary Fig. 7). To correct for this batch

effect, we used monocytes which were included in all three studies as a bridging cell type. This was done by finding the average Beta values across all monocytes in each study separately to create an average monocyte profile for each study. These three average monocyte profiles were further averaged to generate a cross-study aggregate monocyte profile and the difference between the average monocyte profiles from each study and the cross-study aggregate monocyte profile was defined as the correction factor for each study. The correction factor derived from the monocyte signals for each study was then applied to all cell types in that study. After correction, resulting Beta values less than 0 were set to 0 and greater than 1 were set to 1. This batch correction effectively removed sample clustering by study and resulted in a much clearer cell-type clustering (Supplementary Fig. 7).

In generation of the signature matrix, we first excluded purified leukocyte samples that had >10% CpGs missing, and CpGs which were missing in >10% of the samples. As a second filtration step, 235 sex-associated CpGs on autosomal chromosomes[55] and 11,648 CpGs on the X and Y chromosomes were removed to avoid any sex-associated signals. We further filtered CpGs to avoid contaminating signals from nonhematopoietic cell types which was inspired by the construction of the LM22 signature matrix in CIBERSORT[5]. In our case, as we aimed to select for immune cell-type-specific CpGs with low methylation, we removed CpGs with an average Beta <0.8 across a panel of ~800 solid cancer cell lines[56] (GSE68379) in order to mitigate potential contaminating signals from entering the signature matrix (Supplementary Data 11). Additionally, we also excluded 360,000 potential CpGs with low methylation (average Beta <0.8) in any nonhematopoietic tissue based on a set of cell lines derived from many different primary human tissues[57] (GSE31848, GSE59091, GSE68134, Supplementary Data 12).

From the remaining 60,557 candidate CpGs, we conducted t-tests between all pairwise combinations of the 11 immune cell types and an additional pairwise test to distinguish cell types that stem from the same myeloid lineage (monocytes, dendritic cells, and macrophages versus neutrophils and eosinophils). We compiled a list of CpGs for each cell type or groups of cell types which had significant Bonferroni-corrected p-values between that cell type and each of the remaining cell types. To determine the number of CpGs to include in the signature for each cell type, we tested 20–200 CpGs per cell type or groups of cell types from the sorted lists of candidate CpGs. We selected the top 35 CpGs from each cell type and 34 CpGs distinguishing cell types from the same myeloid lineage, that minimized the condition number of the resulting signature matrix, in a manner similar to the construction of LM22 signature matrix in CIBERSORT[5]. The final signature matrix consists of a total of 419 CpGs.

**Determining deconvolution significance threshold**. Although deconvolution methods, regardless of algorithm applied, could produce estimates of cell-type fractions for any given sample, whether a deconvolution is successful or meaningful can be questionable. We consider a successful deconvolution to be one that is conducted on a mixture which contains sufficient quantities of the cell types present in the signature matrix. To define the threshold of a successful deconvolution, we collected 689 true-positive samples from human blood (GSE42861) where a majority of the immune cell types in our reference signature matrix were believed to be present, and 413 true negative samples from normal lung cell lines, non-small cell lung cancer cell lines, breast cancer cell lines, colon cancer cell lines, and ovarian cancer cell lines (GSE36216 and GSE57342), which we did not expect to have significant leukocyte components. We explored four goodness-of-fit metrics, R1, R2, RMSE1 and RMSE2, as detailed below, to evaluate the validity of a deconvolution of these samples.

For each mixture, $m$, we calculated the estimated relative cellular fractions, $\tilde{f}$, using our newly defined robust signature matrix, $B$, and LTS in MethylResolver to solve the equation $m = B \times \tilde{f}$. We then calculated a reconstituted mixture, $\tilde{m}$, by computing $B \times \tilde{f}$. R1 was obtained by calculating the Pearson correlation between our original mixture, $m$, and our reconstituted mixture, $\tilde{m}$, and RMSE1 was calculated similarly by obtaining the root-mean-square error between $m$ and $\tilde{m}$. R2 and RMSE2 were calculated using the best subset of CpGs from the signature matrix as determined by minimizing the residuals in LTS regression, using this subset of CpGs gave the subset signature matrix $\underline{B}$ and subset mixture $\underline{m}$. We then calculated the reconstituted mixture using the best CpG subset, $\underline{\tilde{m}}$, by computing $\underline{B} \times \tilde{f}$. R2 was computed by the Pearson correlation between $\underline{m}$ and $\underline{\tilde{m}}$ and RMSE2 was obtained from the root-mean-square error between $\underline{m}$ and $\underline{\tilde{m}}$.

The best goodness-of fit metric and threshold was determined based on how well the true-positive and true negative samples were separated. We decided to use the R2 goodness-of-fit metric as the significant deconvolution threshold metric as it had the highest discriminatory power in our benchmarking. A significant deconvolution is defined as deconvolution of mixtures achieving a R2 goodness-of-fit value above a pre-defined threshold. To evaluate the performance of this metric on more realistic scenarios, we constructed synthetic mixtures from random pairwise combinations of true-positive and true negative samples. The fraction of unknown content in the mixture was derived from the true negative samples and scaled from 0 to 100% of the total synthetic mixture in increments of 0.1% with 200 random synthetic mixtures at each increment for a total of 200,200 random synthetic mixtures. The synthetic mixtures were deconvolved using MethylResolver and the fraction of significant deconvolutions, which was defined as the percentage of deconvolved mixtures with an R2 value above a given R2 threshold, of the 200 mixtures in each 0.1% unknown content increment, was calculated using a range of

thresholds. A regression line was fit to the percent of significant deconvolutions generated by each threshold using Local Polynomial Regression Fitting with the R function "loess" with a smoothing span of 0.25. This fit line was used to determine the percent of significant deconvolutions at 100% unknown content for different thresholds to obtain an estimate of the false-positive rate.

To benchmark the sensitivity and specificity of our deconvolution threshold, we used the same 200,200 random synthetic mixtures derived from true-positive and true negative samples as described above. Using windows of 5% centered around specific values of unknown content spike-ins (e.g. 27.5–32.5% for 30% unknown content), a total of 10,000 mixtures per window were used as true-positive samples, and 10,000 true negative samples derived from random pairwise mixtures of 519 negative samples (from GSE64511, GSE59091, GSE74877) were used. To generate the ROC curves for each unknown content value, we slid the threshold from 0 to 1 in increments of 0.001.

**TCGA data acquisition**. Level 3 Illumina 450k methylation array Beta values were downloaded from TCGA on March 28th, 2018. We obtained data from 9,756 samples across 33 cancer types in 11 broad categories as defined by TCGA.

**Estimating tumor purity and leukocyte fractions for TCGA**. Various methods have been developed to infer tumor purity, which is known to be an important prognostic indicator. To leverage the strengths of different methods and help reduce noise from any single method, Aran et al. applied four established tumor purity methods to 9364 tumor samples from TCGA. Combining these four methods into a single metric called CPE[23] (consensus measurement of purity estimates) helped improve the accuracy of purity estimates. Here we used the CPE metric as the ground truth tumor purity estimates and trained a random forest (RF) regression model (using R package "randomForest") to predict tumor purity using the four goodness-of-fit metrics: R1, R2, RMSE1, RMSE2, and relative fractions of the 11 leukocyte subsets inferred by MethylResolver. It is important to note that our method is supervised as it was trained on CPE tumor purity measures, whereas previous methods for estimating tumor purity are unsupervised. We first ran MethylResolver on 7001 TCGA samples across 21 cancer types which had both 450k methylation data and CPE tumor purity measures. The RF regression model was built using 500 trees, five features randomly sampled with replacement as candidates at each split, and a minimum of five samples at each terminal node. We trained and tested the model with 5-fold cross validation on half of the available samples from each cancer type (3501 total), selected at random, and evaluated the model on the other half of the samples (3500) which were not seen during training. Additionally, we completely held out all samples from each cancer type in a leave-one-out process during training to get an estimate on the performance of tumor purity estimate in a cancer type that had not been seen at all. To evaluate the performance of the RF model on the held-out testing samples, we compared the tumor purity estimates from our RF model to the ground truth CPE purity estimates. Once the RF model was established, it was applied to estimate tumor purity from MethylResolver deconvolution results on 9442 TCGA samples across 30 different solid tumor types with available level 3 methylation data. These 9442 samples included 3501 samples which were used during the training/ testing of the RF model and 3500 samples (7001 total) that were used in the evaluation of the model. The remaining 2441 samples were not used in either training/testing or evaluation due to missing CPE estimates.

The inferred tumor purity was then used to scale relative leukocyte subset fractions to generate tumor purity-scaled leukocyte subset fractions with the following equation:

$$\tilde{\tilde{f}} = (1-q)*\tilde{f},$$

where $q$ was the estimated tumor purity, $\tilde{f}$ was the relative leukocyte subset fractions, and $\tilde{\tilde{f}}$, was the tumor purity-scaled leukocyte subset fractions.

**Determining significant immune populations in the TME**. To quantify the effect of different immune cell populations on patient outcomes, we applied Cox regressions using the "coxph" function from the 'survival' package in R. Sample-level meta data, including survival times, were obtained from the Cancer Genomics Data Server (CGDS) (http://www.cbioportal.org) using the "cgdsr" package in R. For significant deconvolutions of each cancer type, we constructed a Cox regression model to associate overall survival with tumor purity-scaled or relative cell-type fractions, CD8-to-Treg ratio (CD8/Treg), CD8-to-CD4 ratio (CD8/CD4), or inferred tumor purity, while adjusting for covariates including sex, age, and grade or histology. Whenever available, sex and grade were included in the model as nominal factors, and age as a continuous variable. For the Cox regressions, only tumor samples with significant deconvolutions were used, as we did not want to include lower-confidence results into the regression. Benjamini-Hochberg correction to the resulting p-values was applied to correct for multiple testing. The test used here is more stringent than most other pan-cancer deconvolutions of TCGA tumors which often employed univariate analyses without adjusting for covariates. Only tumor samples with significant deconvolutions (using a deconvolution significance cutoff of R2 = 0.5) were included in our Cox regression. FDR ≤ 0.15 was used as the cutoff for statistical significance similar to a previous study[16]. For the

Kaplan-Meier survival curves, pairwise post hoc analysis for the Logrank-test was used to find significant differences between groups.

**Statistics and reproducibility**. Data in the graphs are expressed as mean ± standard error of mean (SEM). ANOVA post hoc analysis was used to compare the performance of the different deconvolution methods on in silico mixtures. Pearson correlations are primarily used throughout the manuscript, unless Spearman correlations are explicitly specified. Regression lines are fit using loess smoothing. TCGA survival data was compared using post hoc analysis for the Logrank-test and multiple comparisons were performed with a Cox regression. All hypothesis testing was two-sided with a significance level of 0.05. Statistical analyses were done using R, version 3.5.1.

**Reporting summary**. Further information on research design is available in the Nature Research Reporting Summary linked to this article.

## Data availability

Data used to generate the in silico mixtures used in Fig. 1a and Supplementary Fig. 1 and 2a were obtained from GSE88824. The data used to generate the in vitro mixtures used in Fig. 1b and Supplementary Fig. 3a, b and the mixtures used to benchmark the new signature in Fig. 1c and Supplementary Figs. 8a, b and 9 were obtained from GSE77797. Data for the construction of our reference signature was obtained from GSE35069, GSE59250, and GSE71837. Data used to filter the reference signature included a panel of solid cancer cell lines (GSE68379) and cell lines derived from nonhematopoietic human primary tissues (GSE31848, GSE59091, GSE68134). Data used as a true positive in evaluating the detection threshold of MethylResolver was obtained from GSE42861. Data used as true negatives in Fig. 2a, b were obtained from GSE36216 and GSE57342. Data used as true negatives in the mixtures in Fig. 2c, d were obtained from GSE64511, GSE59091, and GSE74877. Data behind Figs. 1, 2, 3, and 5b are available at https://doi.org/10.6084/m9.figshare.12543473[58]. Data behind Fig. 4 are available in Supplementary Data 7 and Supplementary Data 8. Data behind Fig. 5a are available in Supplementary Data 10 and data behind Fig. 5c–e are available in Supplementary Data 8. The data that support the findings of this study are available from TCGA but restrictions apply to the availability of these data, which were used under license for the current study, and so are not publicly available. Data are however available from the authors upon reasonable request and with permission of TCGA. Any data not present in the manuscript or supplementary materials are available from the authors upon reasonable request.

## Code availability

MethylResolver is open source software available on GitHub under the GPLv3 license: (https://github.com/darneson/MethylResolver).

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

## Acknowledgements
The authors thank Jarek Kostrowicki and Keith Ching for insightful and helpful dis-cussions. This work was supported by grants from the NIH-NCI National Cancer Institute (T32CA201160) to DA. Funding for open access charge: National Institutes of Health. The results published here are in whole or part based upon data generated by the TCGA Research Network: https://www.cancer.gov/tcga.

## Author contributions
D.A. and K.W. developed the method, D.A. analyzed the results, and D.A., K.W., and X. Y. wrote the paper.

## Competing interests
K.W. is an employee and shareholder of Bristol-Myers Squibb.
