## [Peer Review File · Communications Biology]

Reviewers' comments:

Reviewer #1 (Remarks to the Author):

The paper MethylResolver by Arneson D et al presents a novel statistical algorithm based on Least Trimmed Squares (LTS) regression, primarily aimed at estimating relative and tumor purity adjusted fractions of immune cell subtypes in complex tissues. As such, the paper addresses an important outstanding challenge, and any novel analysis strategies should be welcomed. However, I found the presentation of the manuscript somewhat unusual, which makes it difficult to read and assess. For instance, the Methods section often refers to Supplementary Figures and Tables, however these should be cited in the Results section. Conversely, when reading the Results section, the authors often withhold key information that is essential for being able to correctly interpret the Figures. Perhaps more worryingly, the manuscript contains a number of statistical procedures which appear ad-hoc and which are not well justified, e.g. the use of β' over β is very unclear. This also applies to some of the metrics used to compare competing methods. The implementation of LTS also involves a parameter, which needs to be optimized, yet the procedure for optimizing it is not clearly presented, raising concerns about overfitting. Below is a list of major concerns:

Major concerns:

1) Metric used in Fig.1a is not exhaustive: I find the metric used in Fig.1a, displayed on the y-axis, inappropriate because the in-silico mixtures are generated by mixing together 6 immune cell subtypes with a tumor epithelial cell-line, yet the cell-type fraction displayed on the y-axis only refers to one of the 6 immune cell subtypes. I don't think that results would differ substantially if the authors were to present a performance metric that evaluates performance in relation to all 7 cell-types, yet I think it is important to check this. Hence, I would advise the authors to use an R^2 and RMSE between estimated and true cell-type fractions, for each of the 7 cell-types separately, and then to average results over cell-types.

2) Leukocyte DNAm reference used in Fig.1: A major concern regarding Fig.1, is that it is unclear which leukocyte DNAm reference was used to run the different methods. I presume that the same reference was used for all methods, and according to the Methods description it would appear that the authors are using the reference from the IDOL paper. Worryingly, from the description in Methods, it would also appear that the in-silico mixtures were generated from the same purified samples used to generate the DNAm reference. This would definitely lead to overfitting and overoptimistic performance values. While this may not affect the comparison between methods, we can't be 100% sure. Hence, the authors may have to redo this analysis, ensuring that the purified blood cell subtype profiles that are mixed together are not taken from the Koestler IDOL paper. There is ample data from purified blood cell subtypes from other studies (Blueprint, Zilbauer) that could be used.

3) β' : The rationale for using $\beta' = 1 - \beta$ is unclear to me. Indeed, I am not convinced at all that using β' should be better, either mathematically, or biologically. For instance, the argument that immune-cell type specific markers are generally hypomethylated, while true, does not demand or require the use of β' , because the statistical inference framework of the algorithms considered is inherently symmetric in relation to β or $1 - \beta$. Methylation data is distributed between 0 and 1, and an algorithm aimed at estimating cell-type fractions should perform the same irrespective of whether we use β or β' . Biologically, the argument that high CpG density promoters become hypermethylated in cancer, while true, is also not a justification for using β' over β . Hypomethylation in cancer is also frequently observed. In summary, the authors have the onus here to provide rigorous and clear statistical arguments as to why the statistical inference is dependent on using β or β' . Adding a figure where the justification of β' is conceptually displayed is important.

4) Construction of more comprehensive leukocyte DNAm reference (line 251): the authors add 34 CpGs to distinguish monocytes, dendritic cells, and macrophages from neutrophils and eosinophils, to their reference. I found this procedure very unclear and ad-hoc, and no justification is provided.

Perhaps more worryingly, the selection of CpGs to make up the reference, starts out from a very small fraction (~70,000) of all CpGs available. The authors are thus using a very small pool of CpGs, which suggests that their leukocyte reference could be far from optimal. The underlying problem here is that the authors are filtering CpGs based on their baseline methylation levels in various cell-types, but the need for this is both unclear and questionable.

5) TCGA analysis (line 322): I don't understand why the authors decided to leave HNSCC out, whilst other cancer-types were not. Should the authors not do the same analysis but now leaving out in turn each one of the other cancer-types?

6) Equation (line 330): I am not sure the equation is correct....should it not be $(1-q)*f$? the tumor-adjusted fractions plus the tumor purity index should add to 1.

7) LTS feature selection procedure is not well described: LTS has a parameter alpha, which controls the fraction of CpGs from the reference that are used in the decomposition. I would like to see a clearer explanation as to how this parameter should be selected, say if we want to apply MethylResolver to some new cancer DNA methylation datasets, where there is no ground truth. I think that not presenting a concrete procedure for estimating this parameter on an arbitrary dataset is a big limitation of the method.

8) Clarity on the threshold for calling a significant deconvolution: Another important parameter in MethylResolver, is the threshold which determines whether a deconvolution is significant or not. How dependent is this threshold on the specific study is unclear, as the authors only assess cancer samples from TCGA. Moreover, the fact that only 34% or so of TCGA samples exhibit a significant deconvolution does highlight a severe limitation of the method, since these cancer samples would contain few immune cells and therefore could be important in a prognostic setting. I therefore worry considerably how useful MethylResolver could really be if it is only applicable to 30% of tumors.

Reviewer #2 (Remarks to the Author):

Comments to the manuscript entitled: "MethylResolver – A Method for Deconvoluting Bulk Tissue DNA Methylation Profiles into Known and Unknown Cell Contents":

Arneson and colleagues present MethylResolver, a regression-based method (Least Trimmed Squares, LTS) for deconvolution of bulk tumor tissue using DNA methylation data to infer relative leukocyte fractions in tumor admixtures using a reference data set consisting of 11 leukocyte cell types. LTS has been shown to be robust to outliers and unknown cellular content, which is relevant in estimating cell types in tumors where the cellular composition is often not as well characterized as for example whole blood. Importantly, MethylResolver can estimate tumor purity and deconvolute infiltrating leukocyte cell type fractions as both relative and absolute cell type fractions using a reference-free approach (random forest regression).

The authors describe a more accurate method compared to existing procedures to resolve unknown cellular content and estimation of cell type fractions without using a cancer-specific reference.

Knowledge about infiltrating leukocytes in tumors is important to reveal cancer progression and response to treatment. In this setting, MethylResolver could represent an important contribution to this field. Overall, the manuscript is well written with a detailed, comprehensive description of the methods and convincing results testing and benchmarking the method and leukocyte reference dataset. However, some clarifications are needed which could help interpretation and validity of the results presented.

Comments:

1. The abstract could benefit from including some of the actual results. As it stands it is not very informative.

2. DNA methylation is given as 1-Beta, which is a measurement of unmethylated DNA. The authors claim that this is more appropriate than 0-1 since immune cell markers are often hypomethylated and hence better for deconvolution purposes in a cancer setting. However, it would be interesting to see results supporting this. The authors argue that this could minimize any influence by unknown tumor content could have on the deconvolution and resulting estimates. However, some results showing that the assumption that unknown tumor content displays a different DNA methylation distribution would be useful. How would this for example affect the tumor purity estimates?
3. To build a comprehensive leukocyte reference dataset, DNA methylation data for 11 cell types from three publicly available studies were included. The data were corrected for batch effects using B cells (included in all data sets) as a bridging cell type. Although this batch correction removed variance related to dataset, it also resulted in overlap of different cell types (Supplementary Fig 4), which could make it difficult to select L-DMR library for deconvolution. Right? I am surprised to see the high (very close to 1) R2 of the resulting estimates versus FACS counts. Do the authors have any thoughts about this? One would assume that highly overlapping cell clusters challenge selection of a library differentiating cell types by IDOL?
4. As the authors describe there are additional leukocytes not included in the extended reference, and future single-cell sequencing experiments will reveal the overall complexity, which could be included. What about other cell types (for example endothelial cells) at this stage?
5. Only validation of 6 cell types are presented, what about the remaining cell types included in the extended reference data set?

Reviewer #3 (Remarks to the Author):

This paper describes a new reference-based deconvolution method, MethylResolver, to estimate leukocyte subset fractions from heterogeneous tumor cell populations using DNA methylation profiles. It estimates both relative fractions as well as fractions scaled by the fraction of tumor cells in the population without needing a tumor reference signature. Furthermore, it provides a measure of goodness-of-fit of the estimated subset fractions to judge whether there is sufficient signal in the data to extract these from the bulk sample. The paper is very well-organized and well-written, except I think it is missing a lot of hyphens (e.g. false-positive rate, 5-year survival, etc.)

General comments:

The name "tumor purity-adjusted leukocyte fractions" is a confusing use of the word "adjusted". Given the formula, I would recommend calling these tumor purity-scaled leukocyte fractions, because you are multiplying the fractions by a scaling factor.

Please delete the word 'multivariate' when used to describe Cox regression. Multivariate would imply there is more than 1 survival time, and there is not. You are just talking about regression with multiple independent variables. This is not multivariate, but multivariable. Nevertheless, most people understand that regression is multivariable and do not say 'multivariable regression'. So, please say "we applied Cox regression...". And "...we constructed a Cox regression model..."

Specific comments:

1. Please add the age and ethnicities of the samples from which the 11 reference leukocyte cell types were measured.
2. Abstract, sentence: "In silico deconvolution offers a fast and inexpensive alternative to inferring

cellular fractions from bulk tissue data." What is In silico deconvolution if not inferring cellular fractions from bulk tissue data?

3. Section: Measurement of methylation levels, line 3, the term "hypomethylated" is a relative term but no reference sample is mentioned. Are you trying to say they have low levels of DNA methylation? Please say this instead of 'hypomethylation'.

4. Section Model formulation, last sentence: "... LTS can remove CpGs which are contaminating outliers and contain signal which was not present in the signature matrix thereby improving the cell fraction estimation." I do not understand "contain signal which was not present in the signature matrix" Please clarify.

5. Section: Building a more comprehensive immune cell type-specific methylation signature matrix: the following sentence is unclear: "The correction factor for each study was then applied to all samples across all cell types in each study" Are you trying to say that the process is repeated for each cell type, and not that the monocyte correction is applied to all cell types? Please clarify.

6. Same section as above, please end the section by giving the total number of CpGs in the signature matrix (Is it $419 = 35 \times 11 + 34$?)

7. Section: Determining deconvolution significance threshold and detection limit, paragraph 3. This paragraph is a little too brief to understand. You use the term "significant deconvolution threshold metric" without defining it. Please define clearly.

8. Same paragraph as in 6.: Please describe/explain "the fraction of significant deconvolutions of the 200 mixtures"

9. Section: Estimating tumor purity...

Paragraph 1, last sentence. Not only does CPE metric benchmark the performance, but you use it to train the model. Previous methods are unsupervised, yours is supervised. Please add this distinction.

10. Section: Estimating tumor purity...

Paragraph 2, sentence 2. Please delete "from matched samples". If I understand this correctly, you don't have pairs of samples that are matched, you have different data types from the same sample (already described as such in sentence 1).

11. Section: Estimating tumor purity...

Paragraph 2, further down. "...it was applied to 9,442 samples across 30 different cancer types..." Please give the overlap between this sample set and the 7001 samples across 12 cancer types used for training.

12. Section: MethylResolver provides statistical assessments for the significance of deconvolution.

Paragraph 1 end. Please give us the number of CpGs determined from the LTS regression model.

Paragraph 2. Please explain a little more about Figure 2c, and how many samples are analyzed for each estimate of Percent significant.

13. Section: Estimating relative and tumor purity-adjusted(scaled) leukocyte fractions

Sentence 2. Please interpret the meaning of "we observed strong correlations between CPE and each of the four goodness-of-fit metrics" and mention again what the goodness of fit is measuring.

14. Section: Application of MethyIResolver for pan-cancer analysis of TCGA

Please give the overlap between this set of 9756 and the 7001 used for training the RF.

Is there a different % significant deconvolutions in the training data set compared to the remainder of tumors?

15. Section: Identification of prognostic leukocyte subsets in TME

Please summarize the similarities and differences you see in Figure 5b and Suppl Figure 13, the results for the purity-scaled subset fractions and the relative subset fractions. What is the effect of scaling on this analysis?

16. Discussion section, paragraph 2. Please explain why your purity-scaled subset fractions are not already an estimate of the absolute fractions. Why is this only a step in that direction?

17. Discussion paragraph 3. "...to discriminate these confounding scenarios...". Please use a word different from confounding. This is not the proper use of this word.

18. Page 18, line 1. "These results give us..." not "results gives us"

Response to Reviewer Comments

We thank the Editor and the three Reviewers for the constructive and thoughtful comments to help us improve the study. We have revised the manuscript to address each of the comments, as outlined below.

Reviewers' comments:

Reviewer #1:

The paper MethylResolver by Arneson D et al presents a novel statistical algorithm based on Least Trimmed Squares (LTS) regression, primarily aimed at estimating relative and tumor purity adjusted fractions of immune cell subtypes in complex tissues. As such, the paper addresses an important outstanding challenge, and any novel analysis strategies should be welcomed. However, I found the presentation of the manuscript somewhat unusual, which makes it difficult to read and assess. For instance, the Methods section often refers to Supplementary Figures and Tables, however these should be cited in the Results section. Conversely, when reading the Results section, the authors often withhold key information that is essential for being able to correctly interpret the Figures. Perhaps more worryingly, the manuscript contains a number of statistical procedures which appear ad-hoc and which are not well justified, e.g. the use of Beta' over Beta is very unclear. This also applies to some of the metrics used to compare competing methods. The implementation of LTS also involves a parameter, which needs to be optimized, yet the procedure for optimizing it is not clearly presented, raising concerns about overfitting. Below is a list of major concerns:

Response: We appreciate the reviewer's constructive feedback on our study. We have reformatted the manuscript as requested and provided detailed responses to the specific critiques raised by the reviewer below.

Major concerns:

1.1 Metric used in Fig.1a is not exhaustive: I find the metric used in Fig.1a, displayed on the y-axis, inappropriate because the in-silico mixtures are generated by mixing together 6 immune cell subtypes with a tumor epithelial cell-line, yet the cell-type fraction displayed on the y-axis only refers to one of the 6 immune cell subtypes. I don't think that results would differ substantially if the authors were to present a performance metric that evaluates performance in relation to all 7 cell-types, yet I think it is important to check this. Hence, I would advise the authors to use an R^2 and RMSE between estimated and true cell-type fractions, for each of the 7 cell-types separately, and then to average results over cell-types.

Response: We appreciate the reviewer's suggestion for evaluating the performance the different methods in **Fig. 1a** and have replaced the figure with one that shows the R^2

(and error bars indicating the SEM) between the estimated and true cell-type fractions for each of the 6 cell types included in the signature. Although there are 7 cell types in the *in silico* mixture, including 6 immune cell types and 1 solid tumor cell line to mimic any unknown content, this solid tumor cell line was not included in the signature matrix used for deconvolution, therefore not appropriate for inference. The goal of MethylResolver is to be able to accurately estimate immune cell type fractions (that are modeled in the signature matrix) in the presence of unknown content in the tissue admixture. While we do introduce a method to infer tumor purity later in the manuscript, the other methods benchmarked here do not directly include such an inference. For this reason and to keep the flow of the paper, in this benchmark we do not infer the cell fraction of the tumor cell lines which are not included in the signature matrix.

We have also included a panel in **Fig. 1a** which gives the R^2 of the aggregate across all 6 cell types. The original **Fig. 1a** which showed how close the estimated cell-type fraction of a given cell type spike-in has been moved to the supplement (**Supplementary Fig. 1**). We substituted the results in **Fig. 1a** with a new *in silico* experiment using independent signature matrices for constructing and deconvoluting the mixtures, as outlined in our response to comment 1.2 below.

1.2 Leukocyte DNAm reference used in Fig.1: A major concern regarding Fig.1, is that it is unclear which leukocyte DNAm reference was used to run the different methods. I presume that the same reference was used for all methods, and according to the Methods description it would appear that the authors are using the reference from the IDOL paper. Worryingly, from the description in Methods, it would also appear that the in-silico mixtures were generated from the same purified samples used to generate the DNAm reference. This would definitely lead to overfitting and overoptimistic performance values. While this may not affect the comparison between methods, we can't be 100% sure. Hence, the authors may have to redo this analysis, ensuring that the purified blood cell subtype profiles that are mixed together are not taken from the Koestler IDOL paper. There is ample data from purified blood cell subtypes from other studies (Blueprint, Zilbauer) that could be used.

Response: We agree with the reviewer's concern. In this revision, we heeded the reviewer's advice and deconvoluted *in silico* mixtures constructed by mathematically combining β' values of from an independent profiling dataset on purified immune cell subsets (**GSE88824**) and a cancer cell line which was absent from the signature matrix. The original IDOL signature was then used for various deconvolution algorithms. These new results are now shown in the updated **Fig 1a** where we evaluated the R^2 as suggested by the reviewer in comment 1.1 and **Supplementary Fig. 1** where we looked at the ability to accurately infer a known cell type spike-in fraction. The new results still substantiate that our method outperforms previous methods.

1.3 β' : The rationale for using $\beta' = 1 - \beta$ is unclear to me. Indeed, I am not convinced at all that using β' should be better, either mathematically, or biologically. For instance, the argument that immune-cell type specific markers

are generally hypomethylated, while true, does not demand or require the use of Beta', because the statistical inference framework of the algorithms considered is inherently symmetric in relation to Beta or 1-Beta. Methylation data is distributed between 0 and 1, and an algorithm aimed at estimating cell-type fractions should perform the same irrespective of whether we use Beta or Beta'. Biologically, the argument that high CpG density promoters become hypermethylated in cancer, while true, is also not a justification for using Beta' over Beta. Hypomethylation in cancer is also frequently observed. In summary, the authors have the onus here to provide rigorous and clear statistical arguments as to why the statistical inference is dependent on using Beta or Beta'. Adding a figure where the justification of Beta' is conceptually displayed is important.

Response: We agree with the reviewer that mathematically Beta and Beta' are symmetrical, and training the method using Beta instead of Beta' would have achieved similar statistical performance. The difference is that different sets of markers will be selected and the biological meaning of the marker sets is not symmetrical between Beta and Beta'. Traditionally (especially in the gene expression field), people often select cell type-specific markers that are highly expressed in a cell type. Such "highly expressed" genes tend to have low promoter methylation, which correspond to large Beta' values. For example, using Beta' we have selected a methylation marker (cg00219921) in the promoter of CD8A as part of the CD8 T-Cell signature, cg10628126 in the promoter of GNLY as part of the NK cell signature, and cg25971649 in an enhancer of CD58, a known cell surface marker of macrophages, as part of the macrophage signature. In contrast, if one selects markers with high Beta values, the resulting markers will likely be lowly expressed genes in the respective cell type. While such markers exist, their biological interpretation may be less intuitive.

We discuss Beta vs Beta' on lines **371-374** as follows:

"Methylation data is typically reported as Beta values bounded between 0 (unmethylated) and 1 (fully methylated). Since cell type specific markers often display high expression in the respective cells, we chose to use Beta', which is calculated as 1-Beta and is a measurement of low levels of DNA methylation with large values likely translating to high cell type-specific gene expression^{47,48}."

1.4 Construction of more comprehensive leukocyte DNAm reference (line 251): the authors add 34 CpGs to distinguish monocytes, dendritic cells, and macrophages from neutrophils and eosinophils, to their reference. I found this procedure very unclear and ad-hoc, and no justification is provided. Perhaps more worryingly, the selection of CpGs to make up the reference, starts out from a very small fraction (~70,000) of all CpGs available. The authors are thus using a very small pool of CpGs, which suggests that their leukocyte reference could be far from optimal. The underlying problem here is that the authors are filtering CpGs based on their baseline methylation levels in various cell-types, but the need for this is both unclear and questionable.

Response: We appreciate the reviewer's feedback and have added clearer explanations of the filtering process. Since the primary application of our algorithm is to deconvolute immune cell subset fractions in solid tumor tissue admixtures, we filtered CpGs to avoid confounding signals from nonhematopoietic cell types following a procedure inspired by the established method CIBERSORT. We first demonstrated that this procedure does not have any detrimental impact on the resulting signature performance in terms of the baseline task of immune cell deconvolution in whole blood samples. This is supported by the fact that our signature (**Fig. 1c**) performs equivalently to the IDOL signature (**Supplementary Fig. 9**), which was constructed from all CpGs without the filtering steps as employed in our approach, in the deconvolution of whole blood samples. One of the important differences between our signature and the IDOL signature is that we employed additional filtering steps to exclude markers bearing Beta' signals from nonhematopoietic cell types, which is critical when it comes to deconvoluting immune cell subsets from tissue admixtures with large unknown content. We have added additional details and clarifications to the signature generation processes on lines **494-513**.

1.5 TCGA analysis (line 322): I don't understand why the authors decided to leave HNSCC out, whilst other cancer-types were not. Should the authors not do the same analysis but now leaving out in turn each one of the other cancer-types?

Response: We selected HNSC as an example to assess the performance of our tumor purity inference procedure in cancer types not seen at all during training of the model. We appreciate the reviewer's suggestion to do this for all cancer types and have conducted the suggested analysis to leave out each cancer type entirely from the training (**Supplementary Fig. 13**). As shown by the new analyses, the performance is largely consistent for most cancer types, except those with very few samples. We have also updated **Fig. 3** where half of all cancer types were held out from training. The updated explanation of these analysis can be found on lines **204-209**.

"To attain performance estimates for tumor purity inference for cancer types not seen in the training, we sequentially held each cancer type out entirely from the training of the model and evaluated the performance on the held-out cancer types (**Supplementary Fig. 13**). Most cancer types achieved a correlation $r > 0.75$ in the leave-one-out analysis indicating a lower bound on tumor purity estimates for cancer types not used in RF model training."

1.6 Equation (line 330): I am not sure the equation is correct....should it not be $(1-q)*f$? the tumor-adjusted fractions plus the tumor purity index should add to 1.

Response: We thank the reviewer for pointing this out and have updated the equation on **line 590**.

$$\tilde{f} = (1 - q) * \tilde{f},$$

1.7 LTS feature selection procedure is not well described: LTS has a parameter alpha, which controls the fraction of CpGs from the reference that are used in the decomposition. I would like to see a clearer explanation as to how this parameter should be selected, say if we want to apply MethylResolver to some new cancer DNA methylation datasets, where there is no ground truth. I think that not presenting a concrete procedure for estimating this parameter on an arbitrary dataset is a big limitation of the method.

Response: In our benchmarking study on the selection of the LTS parameter alpha in **Supplementary Fig. 4** and **Supplementary Fig. 5**, we showed that alpha values between 0.5 and 0.7 achieved similar performance for this particular benchmark. To help users select the alpha values, we implemented a grid search algorithm to automatically identify the alpha value which minimizes the RMSE of the reconstructed immune cell signature profile (based on inferred cell fractions) compared to the original profile if the user does not specify their own alpha value. We have added detailed clarifications of this process in the **Results** on lines **129-131** and in the **Methods** on lines **427-430**.

“To help optimize alpha selection, our MethylResolver implementation empirically tests nine values for alpha from 0.5 to 0.9 in increments of 0.05 and selects the value with the lowest RMSE between the reconstructed immune cell signature profile based on inferred cell fractions and the observed profile, unless users supply their own choice.”

1.8 Clarity on the threshold for calling a significant deconvolution: Another important parameter in MethylResolver, is the threshold which determines whether a deconvolution is significant or not. How dependent is this threshold on the specific study is unclear, as the authors only assess cancer samples from TCGA. Moreover, the fact that only 34% or so of TCGA samples exhibit a significant deconvolution does highlight a severe limitation of the method, since these cancer samples would contain few immune cells and therefore could be important in a prognostic setting. I therefore worry considerably how useful MethylResolver could really be if it is only applicable to 30% of tumors.

Response: We would like to clarify that this significance threshold is an additional confidence measure that reflects the balance between false positives and false negatives, analogous to an FDR assessment. Other tools often lack this measure, making it difficult to evaluate how robust the deconvolution results would be. The addition of this measure to MethylResolver can provide users with a confidence level that avoids over-interpreting the results. This measure, however, does not limit the use of the deconvolution results. Using FDR as an analogy, one may still investigate results that do not meet an FDR cutoff, but the availability of the FDR assessment is an important and valuable statistical feature to help assess how trustworthy the results are. This is discussed on lines **295-299**.

“The significance threshold provides users with a confidence level that avoids overinterpretation of the results, however, it does not limit the use of the deconvolution

results. The threshold can be thought of as analogous to an FDR assessment which provides a metric to quantify the trustworthiness of results but also allows for investigation of said results if they do not meet a predefined cutoff.”

Regarding the fact that only ~30% of TCGA tumor samples produced a significant deconvolution, this parameter can be modified to achieve higher percentages of significant deconvolutions based on the chosen threshold (e.g. 75.8% **Fig. 4a** vs 35.2% in **Fig. 4b**). However, we chose to use a stringent threshold which better balances false positives and false negatives based on our simulation study (**Figure 2c** and **Figure 2d**) which is described in the **Results** on **lines 177-183** and in the **Methods** on **lines 535-557**. We argue that the low fraction of significant deconvolutions is mainly due to the sample collection procedure employed by TCGA studies, which were not designed for the purpose of immunophenotyping. In fact, TCGA studies only included tumors with high tumor cellularity, making it difficult to assess the TME components in general. In support of this, all immune cell deconvolution methods suffer performance drops at very high levels of tumor content in our *in silico* mixture experiment (**Fig. 1a** and **Supplementary Fig. 1**) and such results should be interpreted with caution. Under such circumstances, it is possible that over-interpretation of *in silico* deconvolution results can occur without a confidence measure for the significance of deconvolution. This was precisely what our simulation studies tried to address, through which we determined the optimal cutoff that strikes a good balance between false positives and false negatives.

Reviewer #2 (Remarks to the Author):

Comments to the manuscript entitled: “MethylResolver – A Method for Deconvoluting Bulk Tissue DNA Methylation Profiles into Known and Unknown Cell Contents”:

Arneson and colleagues present MethylResolver, a regression-based method (Least Trimmed Squares, LTS) for deconvolution of bulk tumor tissue using DNA methylation data to infer relative leukocyte fractions in tumor admixtures using a reference data set consisting of 11 leukocyte cell types. LTS has been shown to be robust to outliers and unknown cellular content, which is relevant in estimating cell types in tumors where the cellular composition is often not as well characterized as for example whole blood. Importantly, MethylResolver can estimate tumor purity and deconvolute infiltrating leukocyte cell type fractions as both relative and absolute cell type fractions using a reference-free approach (random forest regression).

The authors describe a more accurate method compared to existing procedures to resolve unknown cellular content and estimation of cell type fractions without using a cancer-specific reference. Knowledge about infiltrating leukocytes in tumors is important to reveal cancer progression and response to treatment. In this setting, MethylResolver could represent an important contribution to this field. Overall, the manuscript is well written with a detailed, comprehensive description of the methods and convincing results testing and benchmarking the method and leukocyte reference dataset. However, some clarifications are

needed which could help interpretation and validity of the results presented.

Comments:

2.1 The abstract could benefit from including some of the actual results. As it stands it is not very informative.

Response: We have revised the abstract to include more specific results.

2.2 DNA methylation is given as 1-Beta, which is a measurement of unmethylated DNA. The authors claim that this is more appropriate than 0-1 since immune cell markers are often hypomethylated and hence better for deconvolution purposes in a cancer setting. However, it would be interesting to see results supporting this. The authors argue that this could minimize any influence by unknown tumor content could have on the deconvolution and resulting estimates. However, some results showing that the assumption that unknown tumor content displays a different DNA methylation distribution would be useful. How would this for example affect the tumor purity estimates?

Response: We appreciate the reviewer's feedback. As detailed in our earlier response to Reviewer 1 comment **1.3**, from a mathematical standpoint, using Beta or Beta' should produce similar deconvolution performance due to their symmetrical relationship. However, we believe that using Beta' offers an interpretation advantage in that cell type specific markers based on Beta', which represent less methylated gene loci, often correspond to genes highly expressed in the corresponding cell type (which is how cell type markers are typically selected). For example, using Beta' we have selected a methylation marker (cg00219921) in the promoter of CD8A as part of the CD8 T-Cell signature, cg10628126 in the promoter of GNLY as part of the NK cell signature, and cg25971649 in an enhancer of CD58, a known cell surface marker of macrophages, as part of the macrophage signature. On the other hand, using the Beta value would select under-expressed genes for a cell type which is less intuitive in interpretation.

We realized that a tumor epigenome can have both hyper- and hypo-methylation events. Our original statement that Beta' may be better when it comes to deconvoluting tumor admixtures was mostly based on knowledge of a cancer-related CpG island methylator phenotype (CIMP) that is characterized by vast hypermethylation of promoter CpG island sites. Cancer cell-specific events such as CIMP will clearly make the use of Beta to identify immune cell specific markers difficult. However, since CIMP does not apply to all cancers, we have removed this argument and would like to focus on the biological interpretability of using Beta' versus Beta.

We discuss Beta vs Beta' on lines **379-382** as follows:

"Methylation data is typically reported as Beta values bounded between 0 (unmethylated) and 1 (fully methylated). Since cell type specific markers often display high expression in the respective cells, we chose to use Beta', which is calculated as 1-

Beta and is a measurement of low levels of DNA methylation with large values likely translating to high cell type-specific gene expression^{47,48}.”

2.3 To build a comprehensive leukocyte reference dataset, DNA methylation data for 11 cell types from three publicly available studies were included. The data were corrected for batch effects using B cells (included in all data sets) as a bridging cell type. Although this batch correction removed variance related to dataset, it also resulted in overlap of different cell types (Supplementary Fig 4), which could make it difficult to select L-DMR library for deconvolution. Right? I am surprised to see the high (very close to 1) R2 of the resulting estimates versus FACS counts. Do the authors have any thoughts about this? One would assume that highly overlapping cell clusters challenge selection of a library differentiating cell types by IDOL?

Response: We thank the reviewer for this question, and it is a very reasonable concern. One of the limitations of using PCA for visualizing our batch correction results is that it is inherently limited in terms of the amount of variance that we can observe in just two principal components. In the original PCA, following batch correction, we found that PC #1 explained most of the variance that separated lymphocytes from non-lymphocytes and that PC #2 separated T cells and B cells. We expect that higher order PCs would start to separate cell types within the T and B cell lineages. To illustrate this point, we used Uniform Manifold Approximation and Projection (UMAP) to embed the top 20 PCs into two dimensions (**Supplementary Fig. 7**) which replaced the original PCA plot (previously **Supplementary Fig. 4**). Using UMAP we are able to better project this high dimensional data into two dimensions and visualize the successful batch correction and the differences between the cell types, which we hope alleviates the reviewer’s concerns. This new visualization is discussed on **lines 481-493** and **lines 491-493**.

“Visualization of the 269 samples from these datasets by embedding of the top 20 principal components in a UMAP (Uniform Manifold Approximation and Projection)⁵⁵ plot using the R package “umap” revealed segregation by dataset (**Supplementary Fig. 7**).....This batch correction effectively removed sample clustering by study and resulted in a much clearer cell-type clustering (**Supplementary Fig. 7**)”

2.4 As the authors describe there are additional leukocytes not included in the extended reference, and future single-cell sequencing experiments will reveal the overall complexity, which could be included. What about other cell types (for example endothelial cells) at this stage?

Response: We thank the reviewer for this question as this is an area of high interest to us as there are great benefits to extending our reference signature. However, in this manuscript, we are limited by the publicly available datasets. We chose to focus on purified cell types which are expected to be present in the tumor microenvironment and had methylation profiling data available. When combining data from multiple studies in the public domain, we were further limited by including only studies with at least one commonly profiled cell type in order to perform batch correction. In future extensions of

this work we would be very interested in including additional cell types such as endothelial cells when appropriate data is available which we discuss on **lines 344-346**.

“In future extensions of this work we would be very interested in including additional cell types such as endothelial cells when appropriate data with overlapping cell types to facilitate batch correction become available.”

2.5. Only validation of 6 cell types are presented, what about the remaining cell types included in the extended reference data set?

Response: We are absolutely interested in validating more immune cells with experimentally confirmed fractions in a tissue mixture. However, such data are currently lacking from public repositories and we only have the publicly available gold standard FACS references for the 6 cell types considered in this work. We discuss the limitation of publicly available gold standard data on **lines 147-150**.

“This evaluation could only be done on 6 cell types in our extended immune cell signature as there are currently no ground truth fractions with matching methylation profiling data available for the remaining cell types.”

Reviewer #3 (Remarks to the Author):

This paper describes a new reference-based deconvolution method, MethylResolver, to estimate leukocyte subset fractions from heterogeneous tumor cell populations using DNA methylation profiles. It estimates both relative fractions as well as fractions scaled by the fraction of tumor cells in the population without needing a tumor reference signature. Furthermore, it provides a measure of goodness-of-fit of the estimated subset fractions to judge whether there is sufficient signal in the data to extract these from the bulk sample. The paper is very well-organized and well-written, except I think it is missing a lot of hyphens (e.g. false-positive rate, 5-year survival, etc.)

Response: We appreciate the positive comments and have added hyphens to places wherever appropriate.

General comments:

The name “tumor purity-adjusted leukocyte fractions” is a confusing use of the word “adjusted”. Given the formula, I would recommend calling these tumor purity-scaled leukocyte fractions, because you are multiplying the fractions by a scaling factor.

Response: We appreciate the reviewer’s suggestion and have used the suggested term of “tumor purity-scaled leukocyte fractions” throughout the revised manuscript.

Please delete the word ‘multivariate’ when used to describe Cox regression.

Multivariate would imply there is more than 1 survival time, and there is not. You are just talking about regression with multiple independent variables. This is not multivariate, but multivariable. Nevertheless, most people understand that regression is multivariable and do not say ‘multivariable regression’. So, please say “we applied Cox regression...”. And “...we constructed a Cox regression model...”

Response: We appreciate the clarification and have made the changes throughout the manuscript as suggested.

Specific comments:

3.1 Please add the age and ethnicities of the samples from which the 11 reference leukocyte cell types were measured.

Response: As suggested, we have added the age and ethnicities of the samples where the information is available (**Supplementary Data 3**).

3.2 Abstract, sentence: “In silico deconvolution offers a fast and inexpensive alternative to inferring cellular fractions from bulk tissue data.” What is In silico deconvolution if not inferring cellular fractions from bulk tissue data?

Response: We have rephrased the sentence on **lines 17-18** in the abstract to clarify what we meant by in silico deconvolution: “*In silico* deconvolution of cellular fractions from bulk tissue data offers a fast and inexpensive alternative to experimentally measuring such fractions.”

3.3 Section: Measurement of methylation levels, line 3, the term “hypomethylated” is a relative term but no reference sample is mentioned. Are you trying to say they have low levels of DNA methylation? Please say this instead of ‘hypomethylation’.

Response: As suggested by the reviewer, we have replaced hypomethylation with “low levels of DNA methylation” on **line 373**.

3.4 Section Model formulation, last sentence: “... LTS can remove CpGs which are contaminating outliers and contain signal which was not present in the signature matrix thereby improving the cell fraction estimation.” I do not understand “contain signal which was not present in the signature matrix” Please clarify.

Response: We have rephrased the sentence to avoid confusion (**lines 442-445**) as follows:

“By subsetting to the k set of CpGs which minimize the sum of squared residuals, LTS can remove CpGs that bear signals from unknown content in the mixture not modeled in

the signature matrix, thereby improving the cell fraction estimation of the modeled cell types.”

3.5 Section: Building a more comprehensive immune cell type-specific methylation signature matrix: the following sentence is unclear: “The correction factor for each study was then applied to all samples across all cell types in each study” Are you trying to say that the process is repeated for each cell type, and not that the monocyte correction is applied to all cell types? Please clarify.

Response: We have clarified the language of the batch correction on **lines 489-491**. We used monocytes which were present across all datasets to identify a batch correction factor for each dataset, which was then applied across all cell types within that dataset.

“The correction factor derived from the monocyte signals for each study was then applied to all cell types in that study.”

3.6 Same section as above, please end the section by giving the total number of CpGs in the signature matrix (Is it $419 = 35 \times 11 + 34$?)

Response: We have listed the total number of CpGs in the signature matrix as requested on **line 513**:

“The final signature matrix consists of a total of 419 CpGs.”

3.7 Section: Determining deconvolution significance threshold and detection limit, paragraph 3. This paragraph is a little too brief to understand. You use the term “significant deconvolution threshold metric” without defining it. Please define clearly.

Response: We have defined what the “significant deconvolution threshold metric” is as requested by the reviewer on **lines 536-538**, as follows:

“We decided to use the R2 goodness-of-fit metric as the significant deconvolution threshold metric as it had the highest discriminatory power in our benchmarking.”

3.8 Same paragraph as in 6.: Please describe/explain “the fraction of significant deconvolutions of the 200 mixtures”

Response: We have added additional details on what “the fraction of significant deconvolutions” is on **lines 544-546**, as follows:

“...the fraction of significant deconvolutions, which was defined as the percentage of deconvolved mixtures with an R2 value above a given R2 threshold...”

**3.9 Section: Estimating tumor purity...
Paragraph 1, last sentence. Not only does CPE metric benchmark the**

performance, but you use it to train the model. Previous methods are unsupervised, yours is supervised. Please add this distinction.

Response: We have added the clarification that our tumor purity estimate is supervised whereas previous methods are unsupervised on **lines 570-572**, as follows:

“It is important to note that our method is supervised as it was trained on CPE tumor purity measures, whereas previous methods for estimating tumor purity are unsupervised.”

3.10 Section: Estimating tumor purity...

Paragraph 2, sentence 2. Please delete “from matched samples”. If I understand this correctly, you don’t have pairs of samples that are matched, you have different data types from the same sample (already described as such in sentence 1).

Response: As suggested, we have deleted “from matched samples”.

3.11 Section: Estimating tumor purity...

Paragraph 2, further down. “...it was applied to 9,442 samples across 30 different cancer types...” Please give the overlap between this sample set and the 7001 samples across 12 cancer types used for training.

Response: We have detailed the overlap between the samples to which we applied our model and the ones used in the training on **lines 581-586**, as follows:

“Once the RF model was established, it was applied to estimate tumor purity using MethylResolver deconvolution results from 9,442 TCGA samples across 30 different solid tumor types with available level 3 methylation data. These 9,442 samples included 3,501 samples which were used during the training/testing of the RF model and 3,500 samples (7,001 total) that were used in the evaluation of the model. The remaining 2,441 samples were not used in either training/testing or evaluation due to missing CPE estimates.”

We would also like to clarify that the RF model for tumor purity inference is a post-processing step based on MethylResolver deconvolution results. MethylResolver itself, including the immune cell subset signature matrix and significant deconvolution cutoff, was not based on any TCGA samples.

3.12 Section: MethylResolver provides statistical assessments for the significance of deconvolution.

Paragraph 1 end. Please give us the number of CpGs determined from the LTS regression model.

Response: We have added the number of CpGs from MethylResolver signature matrix which were used in the LTS regression on **lines 168-171**:

“Although all four goodness-of-fit metrics could distinguish true positive samples from true negative samples with high sensitivity and specificity, the correlation-based metric, R2, which used 210 out of 419 CpGs in our signature matrix determined from LTS regression, performed the best.”

Paragraph 2. Please explain a little more about Figure 2c, and how many samples are analyzed for each estimate of Percent significant.

Response: We have added additional details on **Fig. 2c** and the number of samples included for each point on the plot in the requested section (**lines 172-175**) and the caption of **Fig. 2**.

“To further evaluate the performance of R2 in more realistic complex mixtures, we constructed 200,200 synthetic mixtures by randomly combining pairs of true-positive and true negative samples in various proportions with the true negative percentage ranging from 0-100% in increments of 0.1% with 200 samples per increment.”

Additionally, this information is available in the Methods section under the subsection: “Determining deconvolution significance threshold”, paragraph 3 (**lines 541-543**).

3.13 Section: Estimating relative and tumor purity-adjusted(scaled) leukocyte fractions

Sentence 2. Please interpret the meaning of “we observed strong correlations between CPE and each of the four goodness-of-fit metrics” and mention again what the goodness of fit is measuring.

Response: As suggested by the reviewer, we have reiterated what the goodness-of-fit metric is measuring and added an interpretation of the statement “we observed strong correlations between CPE and each of the four goodness-of-fit metrics” on **lines 191-195**:

“The correlation between CPE and the goodness-of-fit metrics is intuitive as the latter measures how well the reconstructed immune cell signature profile using inferred immune cell fractions recapitulates the observed profile. As tumor purity increases, the inference of non-tumor immune cell fractions becomes more difficult and the reconstructed profile will deviate further from the observed profile.”

3.14 Section: Application of MethylResolver for pan-cancer analysis of TCGA
Please give the overlap between this set of 9756 and the 7001 used for training the RF.

Is there a different % significant deconvolutions in the training data set compared to the remainder of tumors?

Response: The 7,001 TCGA tumor samples used for training/testing and evaluation of the random forest model are a subset of the 9,442 samples for which we estimate tumor purity in the pan-cancer analysis of TCGA (we only estimated purity for solid tumors and excluded 314 hematopoietic cancers (THYM, DLBC, and LAML). Using an R2 threshold of 0.5, we observed 32% significant deconvolutions among the 7,001 training/testing and evaluation samples, and 40% significant deconvolutions in the remaining 2,441 samples due to a lack of CPE estimates. We think these are similar percentages and do not perceive any bias that would be introduced by the availability of CPE estimates.

3.15 Section: Identification of prognostic leukocyte subsets in TME
Please summarize the similarities and differences you see in Figure 5b and Suppl Figure 13, the results for the purity-scaled subset fractions and the relative subset fractions. What is the effect of scaling on this analysis?

Response: We appreciate the reviewer's insightful comment, and have added discussions on the difference between using purity-scaled subset fractions and the relative subset fractions on **lines 254-259**:

“In general, we observed broadly consistent patterns between the prognostic associations generated from the tumor purity-scaled and relative leukocyte subset fractions (agreeing on 21 significant associations out of 25 and 27). However, some of the unique cases found only when using tumor purity-scale leukocyte subsets are supported by literature including improved prognosis due to tissue associate eosinophilia (TATE) in cervical cancers (CESC)²⁸ and better survival outcome in pancreatic cancer (PAAD) treated with chemoradiation with high Eosinophil-to-Lymphocyte Ratio (ELR)²⁹.”

3.16 Discussion section, paragraph 2. Please explain why your purity-scaled subset fractions are not already an estimate of the absolute fractions. Why is this only a step in that direction?

Response: We have added additional clarifications in the discussion as requested on **lines 279-282**:

“We note that tumor purity-scaled fractions are not absolute fractions as we do not currently estimate the contributions of other cell types existing in the tumor microenvironment (e.g. fibroblasts and endothelial cells). Nevertheless, the purity-scaled fractions are closer to the absolute fractions than relative fractions.”

3.17 Discussion paragraph 3. “...to discriminate these confounding scenarios...”. Please use a word different from confounding. This is not the proper use of this word.

Response: We have reworded the phrase as follows: “... to discriminate these unreliable deconvolutions from trustworthy deconvolutions...” on **line 288**.

3.18 Page 18, line 1. “These results give us...” not “results gives us”

Response: We have corrected the typo on **line 318**.

Reviewers' comments:

Reviewer #1 (Remarks to the Author):

The revised paper of Arneson D et al is much improved. I thank the authors for addressing most of my concerns. The only concern that remains however is the issue of using beta over beta'. I read the author's response and the authors seemingly wish to use beta' because they want to select CpGs in regulatory regions (e.g. promoters / enhancers) that are unmethylated because this generally means that the corresponding genes are highly expressed and therefore in line with the notion of what cell-type specific markers should be. I totally agree with this, but my point was then why not select CpGs that have low methylation values (i.e. low BETA values). My point was that because there is an inherent symmetry between beta and beta'=1-beta (since beta-values are bounded between 0 and 1), a mathematical algorithm should not care as to whether you use beta or beta'=1-beta. The same point applies to the feature selection step! Selecting CpGs with large beta' values is equivalent to selecting CpGs with low beta values, right? So, therefore, because this paper is about DNA methylation and NOT gene expression, please revert back to using beta-values. You see, when I first read the manuscript I was extremely confused why the authors were using beta', because in the manuscript you even imply that this "is critical to the good performance of the method".... NO.....what IS critical to the whole procedure is selecting features with LOW methylation in regulatory regions!!! This is what the authors should be emphasizing, and so I would ask the authors to please REMOVE beta' from the whole manuscript. Please reframe in terms of beta. Otherwise there is a risk here of causing unnecessary confusion to those who are not well versed in mathematics and statistics.

Reviewer #3 (Remarks to the Author):

The authors have addressed all but one of the previous comments.

Response 1.3 to Reviewer #1 is not addressed adequately.

Regarding the authors remarks:

"We agree with the reviewer that mathematically Beta and Beta' are symmetrical, and training the method using Beta instead of Beta' would have achieved similar statistical performance. The difference is that different sets of markers will be selected and the biological meaning of the marker sets is not symmetrical between Beta and Beta'...."

My response: Because the measures are symmetrical, you can select low Beta values or high Beta' values, and the same set of markers will be selected. I think you are arguing it would be less biologically meaningful to select high Beta value markers and this is true. However, scientists who study DNA methylation discuss selecting low Beta value markers and do not make the data transformation you have implemented just to say they can pick high Beta' values. Your transformation can add unnecessary confusion to the field. If all results are the same (and you have not shown any reason why they would not be), it is confusing to not use the natural scale and simply select markers with low DNA methylation.

Regarding their additions to the manuscript:

The references cited do not support the authors' remark. Reference 47 does not have any results with gene expression, but only with regard to a cell-type specific DNA methylation markers. Reference 48 is too broad of a review paper. I can accept looking for cell-type specific markers with low DNA

methylation levels, but without a correct reference, I do not accept their assumed relation to gene expression.

Other minor corrections:

- 1.) The last sentence of the abstract is unclear. What is meant by "identifying B cells as a novel negative predictor for papillary renal cell carcinoma"
- 2.) Page 5, line 102: please replace "exiting" with "existing"
- 3.) Page 11, line 335: please replace "gives" with "give"
- 4.) Page 16, lines 499 & 501. $\beta' \geq 0.2$ means $0.8 \geq \beta$. Shouldn't you remove $\beta > 0.8$ instead? If not, more reasoning in the text is needed.

Response to Reviewer Comments

We are pleased that many of the responses we provided in the second revision were satisfactory to the reviewers, and we thank the reviewers for the constructive and thoughtful comments which have helped us further improve our study. We have revised the manuscript to address each of the comments, as outlined below.

Reviewers' comments:

Reviewer #1:

1.1 The revised paper of Arneson D et al is much improved. I thank the authors for addressing most of my concerns. The only concern that remains however is the issue of using beta over beta'. I read the author's response and the authors seemingly wish to use beta' because they want to select CpGs in regulatory regions (e.g. promoters / enhancers) that are unmethylated because this generally means that the corresponding genes are highly expressed and therefore in line with the notion of what cell-type specific markers should be. I totally agree with this, but my point was then why not select CpGs that have low methylation values (i.e. low BETA values). My point was that because there is an inherent symmetry between beta and beta'=1-beta (since beta-values are bounded between 0 and 1), a mathematical algorithm should not care as to whether you use beta or beta'=1-beta. The same point applies to the feature selection step! Selecting CpGs with large beta' values is equivalent to selecting CpGs with low beta values, right? So, therefore, because this paper is about DNA methylation and NOT gene expression, please revert back to using beta-values. You see, when I first read the manuscript I was extremely confused why the authors were using beta', because in the manuscript you even imply that this "is critical to the good performance of the method" NO.....what IS critical to the whole procedure is selecting features with LOW methylation in regulatory regions!!! This is what the authors should be emphasizing, and so I would ask the authors to please REMOVE beta' from the whole manuscript. Please reframe in terms of beta. Otherwise there is a risk here of causing unnecessary confusion to those who are not well versed in mathematics and statistics.

Response: We agree with the reviewer on the potential confusions that Beta' might be causing. As suggested, we have removed Beta' from the entire manuscript and reframed everything in terms of Beta while specifying that we selected features with low methylation levels.

Reviewer #2 (Remarks to the Author):

2.1 I continue to find the results interesting and the manuscript has been substantially improved by the revisions undertaken. However, I still question the

rationale for using Beta-1 in the analyses. I do not want to drag this out, but although it would not change the results in this setting, I am not sure it is better for interpretation purposes. After all, the tumor mixtures can present both hyper- and hypo DNA methylation patterns.

Regarding selection of deconvolution library probes, it would not necessarily be correct to draw parallels to deconvolution using gene expression data in this setting. The authors responded that traditionally such libraries preferentially consist of highly expressed genes and that this corresponds to hypomethylated sites. This is not always the case – and why would this be essential for selecting cell type specific DNA methylation markers? Actually, optimized L-DMRs for whole blood (as an example) consist of both methylated and unmethylated probes.

Response: We agree with the reviewer that introducing the terminology Beta' is potentially confusing. We have instead taken the suggestions from the reviewers to reframe the manuscript in terms of Beta values to conform to conventional terminology.

Reviewer #3 (Remarks to the Author):

3.1 The authors have addressed all but one of the previous comments.

Response 1.3 to Reviewer #1 is not addressed adequately.

Regarding the authors remarks:

“We agree with the reviewer that mathematically Beta and Beta' are symmetrical, and training the method using Beta instead of Beta' would have achieved similar statistical performance. The difference is that different sets of markers will be selected and the biological meaning of the marker sets is not symmetrical between Beta and Beta'....”

My response: Because the measures are symmetrical, you can select low Beta values or high Beta' values, and the same set of markers will be selected. I think you are arguing it would be less biologically meaningful to select high Beta value markers and this is true. However, scientists who study DNA methylation discuss selecting low Beta value markers and do not make the data transformation you have implemented just to say they can pick high Beta' values. Your transformation can add unnecessary confusion to the field. If all results are the same (and you have not shown any reason why they would not be), it is confusing to not use the natural scale and simply select markers with low DNA methylation.

Response: We agree with the reviewer that our use of Beta' is unconventional and may cause confusion to the readers. We have rephrased the manuscript in terms of Beta

and clarified that we selected markers with low methylation levels. This allows us to stick with the conventional use of Beta as the primary metric from a methylation microarray.

Regarding their additions to the manuscript:

The references cited do not support the authors' remark. Reference 47 does not have any results with gene expression, but only with regard to a cell-type specific DNA methylation markers. Reference 48 is too broad of a review paper. I can accept looking for cell-type specific markers with low DNA methylation levels, but without a correct reference, I do not accept their assumed relation to gene expression.

Response: We have replaced the references with those which better support the claim that CpGs with low levels of methylation can indicate cell type specific gene expression:

1. Maruyama et al. – Epigenetic regulation of cell type-specific expression patterns in the human mammary epithelium
2. Zilbauer et al. - Genome-wide methylation analyses of primary human leukocyte subsets identifies functionally important cell-type-specific hypomethylated regions

Other minor corrections:

3.2 The last sentence of the abstract is unclear. What is meant by “identifying B cells as a novel negative predictor for papillary renal cell carcinoma”

Response: We have clarified the language as follows:

“identifying elevated B cell fraction as a novel predictor of poor survival for papillary renal cell carcinoma”

3.3 Page 5, line 102: please replace “exiting” with “existing”

Response: We thank the reviewer for catching this typo and have corrected it in the text.

3.4 Page 11, line 335: please replace “gives” with “give”

Response: We thank the reviewer for catching this typo and have corrected it in the text.

3.5 Page 16, lines 499 & 501. Beta' >= 0.2 means 0.8 >= Beta. Shouldn't you remove Beta > 0.8 instead? If not, more reasoning in the text is needed.

Response: In building our signature, we selected CpGs with low Beta values as cell type-specific markers for the immune cell types of interest. In the filtering step referenced in the comment, we were trying to remove CpGs with low methylation signals (i.e. low Beta values) in solid cancer cell lines or nonhematopoietic tissues as a

way to prevent any contaminating signal from entering our signature. We have made changes on **lines 518-523** in the revised manuscript to hopefully make this point more clear.

“In our case, as we aimed to select for immune cell type-specific CpGs with low methylation, we removed CpGs with an average Beta < 0.8 across a panel of ~800 solid cancer cell lines⁵⁶ (GSE68379) in order to mitigate potential contaminating signals from entering the signature matrix (**Supplementary Data 11**). Additionally, we also excluded 360,000 potential CpGs with low methylation (average Beta <0.8) in any nonhematopoietic tissue based on a set of cell lines derived from many different primary human tissues⁵⁷ (GSE31848, GSE59091, GSE68134, **Supplementary Data 12**).”

REVIEWERS' COMMENTS:

Reviewer #1 (Remarks to the Author):

I have no further concerns

Reviewer #2 (Remarks to the Author):

The authors have addressed my concerns and I have nothing else to add.

Reviewer #3 (Remarks to the Author):

all previous concerns were addressed.